# Hybrid Distributed Optical Fiber Sensor for the Multi-Parameter Measurements

**DOI:** 10.3390/s23167116

**Published:** 2023-08-11

**Authors:** Xiao Zhou, Feng Wang, Chengyu Yang, Zijing Zhang, Yixin Zhang, Xuping Zhang

**Affiliations:** 1Key Laboratory of Intelligent Optical Sensing and Manipulation, Ministry of Education, College of Engineering and Applied Sciences, Nanjing University, Nanjing 210023, China; starryzhou@smail.nju.edu.cn (X.Z.);; 2Shenzhen Research Institute of Nanjing University, Shenzhen 518000, China

**Keywords:** distributed optical fiber sensor, Rayleigh scattering, Brillouin scattering, Raman scattering, optical time domain reflectometry, distributed acoustic sensing, distributed temperature sensing, distributed vibration sensing, multi-parameter measurements

## Abstract

Distributed optical fiber sensors (DOFSs) are a promising technology for their unique advantage of long-distance distributed measurements in industrial applications. In recent years, modern industrial monitoring has called for comprehensive multi-parameter measurements to accurately identify fault events. The hybrid DOFS technology, which combines the Rayleigh, Brillouin, and Raman scattering mechanisms and integrates multiple DOFS systems in a single configuration, has attracted growing attention and has been developed rapidly. Compared to a single DOFS system, the multi-parameter measurements based on hybrid DOFS offer multidimensional valuable information to prevent misjudgments and false alarms. The highly integrated sensing structure enables more efficient and cost-effective monitoring in engineering. This review highlights the latest progress of the hybrid DOFS technology for multi-parameter measurements. The basic principles of the light-scattering-based DOFSs are initially introduced, and then the methods and sensing performances of various techniques are successively described. The challenges and prospects of the hybrid DOFS technology are discussed in the end, aiming to pave the way for a vaster range of applications.

## 1. Introduction

In recent decades, optical fiber sensors (OFSs) have been developed rapidly with the growth of fiber optic communication technology. OFS offers many strengths, such as immunity to electromagnetic interference, high sensitivity, robustness, and flexibility [1,2,3,4,5,6,7,8,9,10]. The application of OFS begins with single-point sensing, which is hard to implement in a long-distance monitoring scenario [11,12,13]. As a comparison, distributed optical fiber sensors (DOFSs) take advantage of the sensing ability of optical fiber, that is, deriving all the information along the fiber. In this case, the entire optical fiber can be regarded as the cascaded independent single-point sensors, revealing the global behavior of a structure rather than extrapolation from a few point measurements [14,15,16]. The unique advantage of long-range distributed measurements has made DOFS a promising technique in industrial monitoring applications.

In general, Rayleigh, Brillouin, and Raman scattering lights are generated in the optical fiber [17]. DOFSs can be classified into various types according to different light scattering mechanisms. When a lightwave with narrow linewidth is injected into the fiber, both the intensity and phase of Rayleigh scattering light are sensitive to external vibration. As the vibration is located through the intensity information, distributed vibration sensing (DVS) is realized [18,19,20,21,22,23]. Later developments showed that the demodulation of the phase will help us to obtain the waveform of the vibration with high definition, which is the basis of distributed acoustic sensing (DAS) [24,25,26,27,28,29,30,31]. DAS has been rapidly commercialized and widely considered in many industrial applications, including pipeline monitoring [32,33,34,35,36,37,38,39], railway and highway transportation [23,40,41,42,43,44,45,46], structural inspection [47,48,49,50], perimeter security [51,52,53,54,55], geophysics [56,57,58,59,60,61,62,63,64,65,66,67,68], and biology and natural sciences [69,70,71]. Brillouin scattering light is characterized by the linear relationship with strain and temperature in the frequency domain [72,73,74,75,76,77]. The Brillouin scattering-based DOFSs are particularly competent in the structural health monitoring of oil and gas facilities [78,79,80,81,82,83,84], power grids [83,85,86,87,88,89,90], tunnels, bridges, and pavements [91,92,93,94,95,96,97]. Meanwhile, geological disaster prewarning [98,99,100,101,102] and geophysical engineering [103,104,105] are also application targets. Moreover, Raman scattering is sensitive to temperature, which leads to the successful development of distributed temperature sensing (DTS) [106,107,108,109,110,111]. Over the past few years, Raman-based DTS has made large progress in fire detection [112,113,114,115,116], pipeline oil and gas industry uses [117,118,119,120,121], hydrological studies [122,123,124,125,126,127], power system monitoring [128,129,130], and dam [131,132,133,134], and tunnel [135,136] structure monitoring. The Raman super-resolved method [137,138,139] is also an attractive tool that is used to study the physical spectral behavior of inspected materials.

However, the variety in the measured parameters (usually temperature, strain, or vibration) is limited no matter which single DOFS is used. In industrial monitoring, there is an urgent demand for collaborative multi-parameter measurement. Typically, pipeline monitoring calls for simultaneous temperature and vibration measurements for the early warning of leakage events [140,141]. The joint real-time monitoring of distributed temperature and cracks in a tunnel is also required to reduce the operating risks of transportation facilities [142,143,144]. As for power transmission lines, the temperature, strain, and vibration are all indispensable parameters for monitoring specific conditions, such as icing and swing [145].

Traditional DOFS only exploits a single scattering mechanism, resulting in limited measurable parameters. Moreover, a single parameter cannot provide enough information about the measurands in many cases, leading to misjudgments and false alarms, which have adverse effects on modern industrial applications. In contrast, multi-parameter measurement brings more valuable information for identifying fault events effectively [146]. Even though it is possible to deploy multiple DOFS devices to get the multi-parameter profiles in a field test, highly increased costs are inevitable. Meanwhile, more than one fiber core will be occupied, and synchronized measurements cannot be guaranteed. Once different sensing mechanisms are combined, and the multiple distributed sensing systems are integrated, it forms a hybrid technique that can realize multi-parameter measurement in an individual DOFS. As a result, the hybrid DOFS technology can not only provide useful multidimensional sensing information, it is also highly efficient and cost-effective.

To combine two or more standalone DOFSs, the compatibilities of the light source, amplifier, detector, methods of modulation, and demodulation should be all considered. Furthermore, it is also necessary to multiplex the system components as much as possible to reduce the overall cost and improve efficiency. In other words, the principles and features of any single DOFS system should be studied, then well-designed methods for integration and optimization could be proposed. In this review, various hybrid DOFSs demonstrated in the past few years will be introduced, and the results of multi-parameter measurements will be exhibited. Concretely, the principle of each DOFS is summarized in Section 2. Then, the advances of hybrid DOFS systems are introduced in Section 3, including the methods used to combine the various DOFS techniques, and the optimization as well. In Section 4, the performance of multi-parameter measurement, which is related to the hybrid DOFSs shown in Section 3, will be demonstrated and described in detail. The challenges and prospects of the hybrid DOFS technology for multi-parameter measurement will be discussed in Section 5.

## 2. DOFSs Based on Light Scattering

As long as a light wave is propagating in the optical fiber, there will be an interaction between the incident light and atoms or molecules of the medium, resulting in the generation of Rayleigh, Brillouin, and Raman scattering lights. Once the surrounding environment (temperature, strain, vibration, or acoustic wave) of the optical fiber changes, the light field will be modulated via the fiber material. As a result, the parameters of the scattering lights (such as intensity, wavelength, frequency, phase, and polarization) will change accordingly. Then the environmental information can be obtained by measuring the variation.

The DOFS schemes include optical time domain reflectometry (OTDR) [147,148], optical frequency domain reflectometry (OFDR) [149,150,151], and optical correlation domain reflectometry (OCDR) [152,153,154]. In general, the OTDR shows the advantage of long-range measurement, and is more often studied in hybrid sensing systems. In this review, all the following mentioned DOFSs are implemented in the time domain.

### 2.1. Rayleigh Scattering-Based DOFS

OTDR is the most well-known Rayleigh scattering-based DOFS. It was proposed by Barnoski et al. in 1976 [155] and has a very simple structure. The basic OTDR configuration is shown in Figure 1. A pulsed fiber laser (common operating wavelength: C-band) generates the probe pulse light, followed by a circulator sending the pulse to the fiber under test (FUT) and directing the backscattered Rayleigh light to the photodetector (PD). The period of the pulse light is longer than its traveling time in the FUT. The location of the Rayleigh scattering single can be determined through the time delay *τ_d_*:(1)$$z=\frac{c\tau_{d}}{2n},$$
where *c* is the velocity of light in a vacuum, and *n* is the refractive index of the fiber. The principle of OTDR is universal in all time domain DOFSs.

In 1982 [156], coherent optical time domain reflectometry (COTDR) was first developed by Healey et al. This can improve the signal-to-noise ratio (SNR) and the dynamic range of OTDR significantly. As depicted in Figure 2, a continuous-wave (CW) fiber laser source (common operating wavelength: C-band) is adopted in COTDR. The coherent detection is realized by mixing the local oscillator (LO) light and the Rayleigh backscattering (RBS) light. An acousto-optic modulator (AOM) is commonly adopted to modulate the probe pulse and provide heterodyne frequency *f*_0_ (from dozens to hundreds of megahertz in general cases). The balanced detection can significantly suppress the direct current (DC) noise, and a polarization scrambler (PS) is necessary to suppress the polarization fading noise. Unlike the broadband light source in OTDR, a narrow linewidth laser is required to guarantee a long coherence length. The electrical fields of the LO light and the probe pulse can be expressed as [25]:(2)$$E_{LO}(t)=\sqrt{P_{LO}}\exp\left(j2\pi \nu_{0}t\right),$$
(3)$$E_{p}(t)=\sqrt{P_{p}}w\left(\frac{t}{\tau }\right)\exp j\left(2\pi \nu_{0}t+2\pi f_{0}t\right),$$
where *P_LO_* and *P_p_* are the power of the LO light and the probe light, respectively, *ν*_0_ is the laser frequency, *τ* is the probe pulse width, and *w* is the window function of the pulse. The photoelectrical field received by the balanced detector (BPD) is the summation of the total Rayleigh scattering signals along the fiber:(4)$$E_{R}(t)=\sum_{i=1}^{N}E_{i}W\left(\frac{t-\tau_{i}}{\tau }\right)\exp j\left[2\pi \nu_{0}\left(t-\tau_{i}\right)+2\pi f_{0}\left(t-\tau_{i}\right)\right],$$
where *N* is the number of the scattering points along the fiber, *τ_i_* is the round-trip time of the *i*th scattering point, and *E_i_* is the corresponding photoelectrical field amplitude. After balanced detection at the BPD, the signal photocurrent can be expressed as:(5)$$\begin{array}{cc} i(t) & \propto ℜ\left\{E_{R}\left(t\right)\cdot E_{LO}^{*}\left(t\right)\right\}\\ & =\sum_{i=1}^{N}A_{i}w\left(\frac{t-\tau_{i}}{\tau }\right)\cos\left[2\pi f_{0}\left(t-\tau_{i}\right)-2\pi \nu_{0}\tau_{i}\right], \end{array}$$
where *ℜ*{**E**} represents the real part of complex **E**, * represents the conjugation operation, and *A_i_* is the photocurrent amplitude converted from the Rayleigh scattering signal of the *i*th scattering point.

For a laser source with narrow linewidth, coherent superposition between the scattering signal is inevitable. This effect is the basis of the phase optical time domain reflectometry (Φ-OTDR) [157]. The structure of the Φ-OTDR setup is similar to those of OTDR and COTDR, which can be regarded as direct detection Φ-OTDR and coherent detection Φ-OTDR (only remove the PS), respectively. Assuming the length of the FUT is *L* and the pulse width is Δ*L*, we divide the FUT into *N* sections by the pulse width. The electric field of the Rayleigh scattering light from the *i*th section can be expressed as [158]:(6)$$E^{i}=E_{0}\exp\left(-2\alpha L_{i}\right)\sum_{k=1}^{M}r_{k}^{i}\exp\left(j\varphi_{k}^{i}\right),$$
where *E*_0_ is the electric field amplitude of the incident light, *α* is the attenuation coefficient of the fiber, *M* is the number of the independent Rayleigh scattering points in the section, *L_i_* is the distance between the beginning of the fiber to the *i*th section (*L_i_* = *i*Δ*L*, i = 1, 2, …, *N*), and *r_k_^i^* and *φ_k_^i^* are the scattering amplitude and phase of the *k*th scattering point in the *i*th section, respectively.

Equations (5) and (6) show that the intensity of Rayleigh scattering is related to both the amplitude and phase of the individual scattering point, and since the scattering points are distributed randomly within the fiber, there is a jagged profile of the Φ-OTDR trace. Whenever an external perturbation is applied to the optical fiber, the Φ-OTDR trace at the corresponding position will change due to the intensity and phase variation. The location of the event can be identified by applying the differential method, in which consecutive temporal Φ-OTDR traces are subtracted from an initial reference one [159]. Although the simplest implementation of Φ-OTDR utilizes the intensity to locate the perturbation, the intensity of the Rayleigh scattering light is nonlinear with the strain induced by the perturbation. On the other hand, the strain on a length of *l* is proportional to the phase change, which can be indicated as [160]:(7)$$\varphi (t,l)=\zeta k\int_{0}^{l}\varepsilon (t,l)dl,$$
where *ε*(*t*, *l*) is the fiber strain and *ζ* = 1 − *n*^2^[(1 − *μ_p_*)*p*_12_ − *μ_p_ p*_11_]/2, *μ_p_* represents Poisson’s ratio, *p*_11_ and *p*_12_ are the components of the strain-optic tensor [161], and *k* = 2*πn*/*λ* is the light propagation constant in the fiber. When a single-mode optical fiber is in use, *ζ* ≈ 0.79.

To quantify the strain of the fiber, there are two major methods, including the extraction of phase and the demodulation of Rayleigh frequency. For phase demodulation, coherent detection is the most often adopted technique, which can about bring higher SNR at the same time [20,162]. Methods such as I/Q demodulation [163,164] and Hilbert transform [165] can be adopted to process the Rayleigh coherent signal. Some direct-detection setups can also derive the phase change. Self-interference methods such as symmetric 3 × 3 coupler [166], phase generated carrier (PGC) demodulation [167], or differentiate-and-cross-multiply [168] methods are applicative. The dual probe pulses [169,170] with different frequency components can also realize the interference, and this is also based on a direct-detection configuration.

Rayleigh frequency demodulation is another method used to retrieve the strain, possessing the advantages of immunity to coherent fading [171] and large strain measurement [25]. Generally, the Rayleigh trace is acquired for different laser frequencies within a given frequency range. Then, the intensity of Rayleigh scattering at each position is observed as a function of frequency. Once there is any strain or temperature change, the intensity–frequency relationship will change due to the variation of fiber refractive index and fiber length. However, one can still recover the relationship by shifting the initial frequency by a certain value. This is a common feature for all kinds of Rayleigh scattering-based sensors, which are capable of providing quasi-static measurements in frequency-sweeping schemes [172]. Algorithms such as cross-correlation [173,174] or least mean squares [175] can be applied to determine the frequency shift, which is proportional to the variations of strain and temperature. The strain and temperature proportionality originates from the thermo-optic and elasto-optic effect of silica, and is similar to the coefficients of the fiber Bragg gratings (FBG) response, which can be denoted as [173]:(8)$$\frac{\Delta \nu }{\nu_{0}}\approx -0.78\cdot \Delta \varepsilon ,$$
(9)$$\frac{\Delta \nu }{\nu_{0}}\approx -\left(6.92\times 10^{-6}\right)\cdot \Delta T,$$
where Δ*ν* is the frequency shift in the spectrum, and Δ*ε* and Δ*T* are strain and temperature change, respectively. Since the frequency-sweeping and the cross-correlation processes are time-consuming, an optimized solution is to employ chirped pulse [176,177]. The chirped pulse method can retrieve the Rayleigh spectrum by transforming the spectral shift to the time delay of the probe, making it possible to implement dynamic measurements.

### 2.2. Brillouin Scattering-Based DOFS

Brillouin scattering is the result of the interaction between incident light and thermally generated acoustic phonons. The propagating phonons form a diffraction grating moving at the velocity of sound, which leads to the frequency shift of the scattering light due to the Doppler effect [15]. The up-shifted and down-shifted Brillouin scattering lights are known as anti-Stokes light and Stokes light, respectively. Considering that the Brillouin scattering light is guided backward in the fiber, the Brillouin frequency shift (BFS) *ν_B_* is determined by the properties of fiber and incident light [17]:(10)$$\nu_{B}=\frac{2nV_{a}}{\lambda_{P}},$$
where *V_a_* is the velocity of the acoustic wave, and *λ_P_* is the wavelength of the pump light in a vacuum. Since the refractive index of fiber is affected by the thermo-optic and elasto-optic effects, BFS has a linear relationship with the change of temperature and strain [178]:(11)$$\nu_{B}\left(T,\varepsilon \right)=\nu_{B}\left(T_{0},\varepsilon_{0}\right)+C_{T}^{B}\left(T-T_{0}\right)+C_{\varepsilon }^{B}\left(\varepsilon -\varepsilon_{0}\right),$$
where *T*_0_ and *ε*_0_ are the initial temperature and strain, respectively, while *C_T_^B^* and *C_ε_^B^* are the temperature and strain coefficients of BFS, respectively. Apparently, it is an intrinsic challenge to discriminate temperature and strain from BFS, which is regarded as the temperature/strain cross-sensitivity [179,180]. Considering that phonons suffer from attenuation in the optical fiber, Brillouin scattering shows a broadened spectrum named Brillouin gain spectrum (BGS), which resembles the Lorentzian curve [181]:(12)$$g_{B}(\nu )=g_{0}\frac{{\left(\Delta \nu_{B}/2\right)}^{2}}{{\left(\nu -\nu_{B}\right)}^{2}+{\left(\Delta \nu_{B}/2\right)}^{2}},$$
where *g*_0_ is the peak gain of the Brillouin scattering, and Δ*ν_B_* is the linewidth of the BGS (from dozens to hundreds of megahertz in general cases).

According to the power of the incident light, Brillouin scattering is further classified into spontaneous Brillouin scattering (SpBS) and stimulated Brillouin scattering (SBS). When the power of the incident light is relatively low, SpBS occurs, which has a linear relationship with the incident power. Once the incident power exceeds a threshold value, SBS occurs, resulting in a significant energy transfer between the incident light and the Brillouin Stokes light. In 1989, the strain and temperature measurements were first implemented with the assistance of Brillouin scattering [77]. Later, the BOTDR and Brillouin optical time domain analysis (BOTDA) sensing schemes were developed [182,183].

A typical BOTDR configuration is exhibited in Figure 3, similar to the COTDR configuration described in Figure 2. The fiber laser is also operated at the C-band, and the linewidth should be less than the spectrum width of BGS by as much as possible. The difference is that the electro-optic modulator (EOM) is usually used to generate the pulse light. In this setup, heterodyne frequency is induced by the BFS *ν_B_*. Similarly to COTDR, a PS is indispensable when suppressing the polarization fading noise. To further enhance the SNR, multiple traces are needed with a certain average time. The beat signal is first converted to a microwave signal (~11 GHz) by a high-speed photodetector. Then, the signal is received by a microwave receiver, which can scan all the frequency components of the microwave signal. Then, various methods such as Lorentz fitting [184,185], quadratic least-square fitting [186,187], the cross-correlation algorithm [188,189], and the cross recurrence plot analysis (CRPA) method [190] are able to acquire the value of BFS. BOTDR has the advantage of single-end access to the FUT, making it easy to implement in engineering.

Since BOTDR utilizes low-power spontaneous light for sensing, the signal has a limited SNR and would immensely restrict the sensing range. The BOTDA configuration, which uses the SBS light, has a competitive advantage related to its higher SNR. As illustrated in Figure 4, the laser source is split into two beams; one acts as the pump pulse light, and the other is modulated into a frequency-shifted probe light. As long as the frequency shift *ν_s_* is near the BFS *ν_B_* of the FUT, there will be an energy transfer between the pump light and the probe light. It should be noted that the EOM in the upper branch outputs a double-sideband probe light, so a narrow bandwidth band-pass filter, whose center wavelength should match the BFS and the operating wavelength of the laser source, is necessary to fetch the single-sideband response. By sweeping the shifted frequency *ν_s_*, the whole range of BGS can be reconstructed. While BOTDA is an excellent choice to extend the total sensing range, both ends of the FUT must be connected to the setup, which makes it challenging to implement in long-range measurements.

### 2.3. Raman Scattering-Based DOFS

During the process of light scattering, a small fraction of the scattering light is scattered by material excitation, and the frequency of the scattered photons is different from that of the incident photons. The two-state transition process can either produce or absorb photons, generating the frequency upshifted anti-Stokes light or the downshifted Stokes light. As a whole, the vibrations of the thermally driven molecules give rise to spontaneous Raman scattering (SpRS) [181]. There is a strong dependence on the temperature of the intensity of the anti-Stokes SpRS, whereas the Stokes SpRS is slightly temperature-sensitive. Such a temperature-dependent characteristic is suitable for distributed temperature sensing.

The first Raman scattering-based DOFS was proposed and demonstrated in 1985 [108], in which the OTDR technique was combined with the SpRS light. Figure 5 gives the typical configuration of ROTDR. The fiber laser commonly works at the C-band. Since the power of the SpRS light is very low (approximately 20–30 dB weaker than the power of Rayleigh scattering light), a probe pulse with higher peak power is necessary. In the detection of the Raman signals, the wavelength-division multiplexer (WDM) is employed to separate the Raman Stokes and anti-Stokes lights. In this case, the WDM should contain channels of 1450 nm, 1550 nm, and 1650 nm. To further increase the detectable level of the weak SpRS light, the avalanche photodiode (APD) with a large gain is generally used. The power of the detected Raman Stokes signal *P_s_*(*z*) and anti-Stokes signal *P_as_*(*z*) as a function of position *z* can be expressed as [111]:(13)$$P_{s}(z)=R_{s}(z)\exp\left[-\left(\alpha_{p}+\alpha_{s}\right)z\right]P_{0},$$
(14)$$P_{as}(z)=R_{as}(z)\exp\left[-\left(\alpha_{p}+\alpha_{as}\right)z\right]P_{0},$$
where *R_s_*(*z*) and *R_as_*(*z*) are the distribution-related coefficients of the Raman Stokes and anti-Stokes lights at position *z*, respectively, while *α_p_*, *α_s_*, and *α_as_* are the attenuation coefficients of the pump, Raman Stokes and anti-Stokes lights, respectively, and *P*_0_ is the laser power. According to Bose–Einstein statistics, *R_s_*(*z*) and *R_as_*(*z*) are approximate to [106]:(15)$$R_{s}(z)≅\frac{1}{\lambda_{s}^{4}}\frac{1}{1-\exp\left[-\frac{hc\Delta \nu_{R}}{k_{B}T(z)}\right]},$$
(16)$$R_{as}(z)≅\frac{1}{\lambda_{as}^{4}}\frac{1}{\exp\left[\frac{hc\Delta \nu_{R}}{k_{B}T(z)}\right]-1},$$
where *λ_s_* and *λ_as_* are the wavelengths of the Raman Stokes and anti-Stokes lights, respectively, *h* is Planck’s constant, *k_B_* is Boltzmann’s constant, Δ*ν_R_* is the Raman frequency shift, and *T*(*z*) is the temperature at position *z*. Then, the ratio of the power of the Raman anti-Stokes and Stokes lights can be calculated as:(17)$$R(z)=\frac{P_{as}(z)}{P_{s}(z)}={\left(\frac{\lambda_{as}}{\lambda_{s}}\right)}^{4}\exp\left[-\left(\alpha_{s}+\alpha_{as}\right)z\right]\exp\left[-\frac{hc\Delta \nu_{R}}{k_{B}T(z)}\right].$$

The above formula indicates that the ratio depends on the statistical temperature distribution *T*(*z*), which is the basis for the distributed temperature sensing of ROTDR. To avoid the error introduced by the environmental change, a reference temperature *T*(*z*_0_) at position *z*_0_ is utilized for calibration, hence the actual temperature profile could be obtained by the following equation [155]:(18)$$\frac{1}{T(z)}-\frac{1}{T(z_{0})}=-\frac{k_{B}}{hc\Delta \nu_{R}}\left[\ln\frac{P_{as}(z)}{P_{s}(z)}-\ln\frac{P_{as}(z_{0})}{P_{s}(z_{0})}+\left(\alpha_{as}-\alpha_{s}\right)\left(z-z_{0}\right)\right].$$

While the pump power should be increased to improve the SNR, nonlinear effects would appear as the power exceeds the threshold, leading to the distortion of the Raman trace. In this case, multi-mode fiber is a common choice in seeking to attain a higher allowable power. However, the intermodal dispersion is detrimental to the sensing range, and the pump pulse would undergo broadening while propagating in the fiber, worsening the achievable spatial resolution [191]. In general, there is a tradeoff between the SNR-related sensing distance and the spatial resolution in ROTDR. The common sensing distance can reach up to few tens of kilometers when the spatial resolution and the measuring time are both within a reasonable range.

In this section, standalone DOFS systems that form the hybrid DOFS technology are mainly reviewed. Basic sensor structures, operating principles, as well as the demodulation methods of the current DOFS are briefly introduced.

## 3. Advances of Hybrid DOFS

Since the DOFSs are classified by different scattering mechanisms, it is essential to combine the Rayleigh, Brillouin, or Raman scattering lights to implement multi-parameter measurements. In recent years, breakthroughs have been achieved in demonstrating the hybrid DOFSs. Not only have methods integrating discrete DOFSs been proposed, but the performance and cost have also been considered.

### 3.1. Combination of Rayleigh and Brillouin Scattering Lights

Among the various hybrid sensing systems, the combination of Rayleigh and Brillouin scattering lights has been the most widely studied in recent years. Both Φ-OTDR and COTDR take advantage of the Rayleigh scattering, and BOTDR and BOTDA use SpBS or SBS light for temperature and strain measurements. Apart from the multi-parameter measurement (temperature, strain, vibration, and loss) based on the integration of discrete sensing characteristics [145,192,193,194,195,196], there is also a focus on the handling of the temperature/strain cross-sensitivity among the Brillouin scattering-based sensors with the assistance of Rayleigh scattering [172,197,198]. In addition, the combination of BOTDA/BOTDR and frequency-sweeping Rayleigh scattering-based DOFS can provide comprehensive and accurate strain or temperature measurements [199,200,201].

When constructing a hybrid Rayleigh and Brillouin sensing system, the main aim is to detect the Rayleigh and the Brillouin scattering lights correctly, especially considering their ultra-narrow frequency interval. In 2020, Wang et al. demonstrated a hybrid sensing system combining Φ-OTDR and BOTDA by using the same set of frequency-scanning optical pulses [199]. The Φ-OTDR part worked in a frequency detuning mode to generate the correlation peak, which is linear with the temperature and strain variations. Since the BOTDA part intrinsically requires a frequency scanning process, the pump pulses were shared for both Φ-OTDR and BOTDA. As shown in Figure 6a, the probe wave for BOTDA was injected from the other end of the sensing fiber. It traveled in the same direction as the Rayleigh scattering light. In this work, fiber Bragg grating (FBG) was utilized to reflect the Rayleigh scattering light and transmit the Brillouin probe light, as exhibited in Figure 6b. In the implemented experimental setup shown in Figure 7, the hybrid system shared the same pulse modulation path, which included an EOM1 to modulate the frequency-scanning signal and an EOM2 to subsequently generate pulses with each frequency component in one temporal frame. Benefitting from the frequency-agile technique [202], this system could conduct dynamic strain measurements.

Another hybrid Φ-OTDR/BOTDA system was proposed by Coscetta et al. in 2021 [192]. It adopted a similar method to separate the Rayleigh and Brillouin signals by the FBG. Figure 8 gives the experimental setup of the integrated system. The lower branch served as the pump pulse for both Φ-OTDR and BOTDA, while the upper branch was the probe light only for BOTDA. The backscattering light was first sent to a narrowband (~5 GHz) FBG to reflect the Brillouin Stokes light. Meanwhile, the Rayleigh scattering light and the Brillouin anti-Stokes light passed through the FBG. Another FBG was tuned only to reflect the Rayleigh scattering light to remove the anti-Stokes component. Finally, the Rayleigh and Brillouin signals are separated and simultaneously detected by two PDs.

In 2016, Zhang et al. presented a pulse modulation method to integrate Φ-OTDR and BOTDR [193]. The experimental setup is illustrated in Figure 9a. An AOM, driven by an arbitrary waveform generator (AWG), was used to generate the modulated pulses. Simultaneously, the optical switch (OS) was triggered by the AWG. By switching the on–off state of the OS, the sensing system acted as the coherent detection BOTDR or the direct detection Φ-OTDR. Even if the Φ-OTDR and the BOTDR could not be implemented synchronously, the well-designed modulated pulses improved the performance of the standalone Φ-OTDR or BOTDR. Figure 9b shows that each modulated pulse comprised several wide pulses *I*_1_ with high intensity and a narrow pulse *I*_2_ with low intensity. The wide pulses were beneficial for the intensity-based Φ-OTDR signal because it needs more energy to excite enough Rayleigh scattering signal and detect perturbations with high frequency. The narrow pulse was for the BOTDR, which requires lower pump power to avoid the nonlinear effects. In addition, the respective spatial resolution was adjustable in the hybrid system.

To further simplify the structure of the hybrid system while maintaining the sensing performance, a hybrid Φ-OTDR/BOTDR setup was demonstrated in 2022 [145]. Since the quantitative demodulation of the Rayleigh scattering phase is crucial for comprehensive vibration measurement, and heterodyne detection is the most effective method to collect the SpBS signal, a double heterodyne detection configuration was proposed. As shown in Figure 10, both the LO path and the scattering light were divided by couplers for independent heterodyne interferences. In this case, the polarization fading noise in BOTDR was suppressed with the help of PS, while the polarization state of the RBS signal remained unscrambled, ensuring the dynamic measurement ability of the Φ-OTDR sub-system. In the meantime, the frequency crosstalk induced by different heterodyne frequencies in Φ-OTDR (AOM frequency shift ranges from tens to hundreds MHz) and BOTDR (Brillouin frequency shift of ~10.8 GHz) was avoided. Most components in this experimental setup were shared, and the overall complexity has been significantly reduced.

Apart from the direct separation of the Rayleigh and Brillouin components in the scattering light, multiplexing is an alternative way to obtain different sensing signals simultaneously. In 2017, Dang et al. employed both the Φ-OTDR and BOTDA through space-division multiplexing (SDM) based on the multi-core fiber (MCF) [200]. From Figure 11, we can see that the continuous light was initially modulated into pulses in the upper branch. A Mach–Zehnder modulator (MZM1) successively shifted the optical frequency for the frequency-scanning Φ-OTDR configuration. Then the other MZM2 shifted the pulse frequency by approximately BFS for the BOTDA function. These two sets of pulses were sent into different cores of MCF through a fan-in coupler. Therefore, the Rayleigh and the Brillouin scattering lights were generated in the corresponding cores and detected independently.

In 2018, Fu et al. demonstrated another hybrid Φ-OTDR/BOTDA system, implementing the wavelength-division multiplexing technology [194]. As depicted in Figure 12, the Φ-OTDR and the BOTDA system conducted measurements from the same fiber segment and shared the distributed amplification components. Signals corresponding to the Φ-OTDR and BOTDA sub-systems were combined and divided by the dense WDM (DWDM). The difference in the center wavelengths of the Φ-OTDR and BOTDA signals should be large enough that the DWDM can filter out each waveband. Besides this, the combination of distributed Raman amplification, distributed Brillouin amplification, and random fiber lasing amplification was employed to extend the sensing range to 150.62 km. The spatial resolutions of Φ-OTDR and BOTDA were 9 m and 30 m, respectively.

On the other hand, BOTDA intrinsically requires the connection of both ends of the fiber, placing restrictions on its implementation in engineering applications. In 2013, Zhang et al. proposed a single-end-access BOTDA integrated with COTDR [195]. The experimental setup is exhibited in Figure 13. There was a short jumper with the FC/PC end-face spliced at the end of the FUT to enhance the Fresnel reflection. Hence, the reflective probe light interacted with the counter-propagating pump pulse to generate SBS. Subsequently, both the reflective Brillouin probe light and the Rayleigh scattering light were mixed with the LO light and detected by a single photodetector. The electric spectrum analyzer (ESA) was used to extract the frequency components of BOTDA and COTDR sensing signals.

In addition to the multi-parameter sensing of the hybrid Rayleigh and Brillouin sensors, the temperature–strain cross-sensitivity problem of BFS was also addressed with the combination of Rayleigh and Brillouin scattering lights. In 2014, Kishida et al. demonstrated a hybrid sensing system based on BOTDA and tunable-wavelength coherent optical time domain reflectometry (TW-COTDR) [197]. Discrete laser sources and receivers were adopted for the BOTDA and the COTDR, as illustrated in Figure 14. Since the Rayleigh frequency shift from cross-correlation and the BFS both have linear relationships to the temperature and the strain but with different coefficients, the temperature and strain could be separated by combining these relationships (Equations (8), (9) and (11)). There comprise the additional benefits derived when combining Brillouin and Rayleigh sensing signals. The Brillouin data were used for distance compensation and thus improved the correlation analysis. On the other hand, the Rayleigh frequency shift could filter the measured BFS and improve the accuracy of the separated strain and temperature.

Recently, Murray et al. introduced another hybrid Rayleigh and Brillouin-based sensing system enabling temperature–strain discrimination [172]. The distinct dependencies of Rayleigh and Brillouin frequency shift on the temperature and strain are devoted to solving the cross-sensitivity problem, as indicated in Equations (8), (9) and (11). To further improve the performance, a slope-assisted BOTDA architecture, which is commonly used for dynamic sensing [203,204], is combined with a frequency-scanning OTDR scheme for quasi-static measurements. Figure 15 depicts the experimental setup of the hybrid system. The laser sources for Rayleigh and Brillouin sub-systems are operated at different wavelengths, and they could be easily separated by a WDM. Specifically, probes of the Brillouin sub-system are modulated by two EOMs with different modulation frequencies. The BOTDA probes consist of four frequency components, including the Brillouin Stokes, anti-Stokes probes, and Stokes and anti-Stokes LOs. The interference occurs at each side in the frequency domain, leaving the Stokes and anti-Stokes interference signals isolated by another WDM, and gathered respectively by two PDs. On the other hand, an AOM pair generates pulses with continuously scanned frequency for the Rayleigh sub-system. Since each AOM is driven at frequencies from 170 MHz to 225 MHz with a 1 MHz step, a train of pulses over a range of 110 MHz with a 2 MHz step is available. In addition, both of the pulses used in Rayleigh and Brillouin sub-systems are set at a width of 50 ns and a repetition rate of 200 kHz, and they are simultaneously directed into the shared FUT.

In 2021, Clément et al. presented an alternative method to solve the cross-sensitivity effect based on a hybrid BOTDR and COTDR system [198]. Figure 16a shows the experimental setup, which uses only one laser source and a common acquisition system. With the help of frequency modulation towards the optical path, multiple frequency components were generated in the backscattering light. Provided the frequency shifts were well-designed, only the Rayleigh and Brillouin anti-Stokes components were retained, leaving the rest outside the 1 GHz bandwidth of the photodetector. Figure 16b indicates the spectral information when the temperature changes. Both the intensity and the center frequency of the Brillouin sensing signal will vary, and the Rayleigh component stays still. As long as the Rayleigh and Brillouin sensing signals were simultaneously collected, the Landau–Plazek ratio could be calculated by the following equation [205]:(19)$$R_{LP}=\frac{I_{R}}{I_{B}}=\frac{T_{f}}{T}\left(\rho_{0}V_{A}^{2}\beta_{T}-1\right),$$
where *I_R_* and *I_B_* are the intensities of Rayleigh and Brillouin signals, respectively; *ρ*_0_ is the material density; *V_A_* is the velocity of the acoustic wave; *β_T_* is the isothermal compressibility at the fictive temperature *T_f_*, and *T* is the current temperature. As the Landau–Plazek ratio was temperature-dependent and related to the intensity only, the strain-induced Brillouin frequency shift was then concluded. Thus, the temperature and strain could eventually be discriminated.

### 3.2. Combination of Rayleigh and Raman Scattering Lights

The Raman-based DOFS, especially the ROTDR, has been widely used for distributed temperature sensing over long distances with meter-scale spatial resolutions. In the meantime, Φ-OTDR is a promising technique for distributed acoustic sensing. Combining the DTS and DAS provides a promising method to measure the temperature and vibration simultaneously for monitoring the leakage events in the oil and gas industry.

Since the wavelength interval between Rayleigh and Raman scattering lights is large, it is easy to separate them with an optical filter or a DWDM. In 2016, Muanenda et al. proposed a distributed acoustic and temperature sensor based on a hybrid Φ-OTDR/ROTDR system [205]. As shown in Figure 17, a commercial off-the-shelf distributed feedback (DFB) laser and a direct detection configuration were adopted. The performance of the laser was thoroughly analyzed and optimized so that the inter-pulse incoherence and intra-pulse coherence occur when using cyclic Simplex pulse coding. The linewidth of the laser was ~4 MHz, and the maximum codeword length was 255, providing a coding gain of up to 9 dB. A high-extinction-ratio Raman filter was used to separate the Rayleigh, Raman Stokes, and Raman anti-Stokes components, which were detected and processed to extract the vibration and temperature information.

To further enhance the SNR and determine the vibration quantitatively, Zhang et al. proposed a hybrid heterodyne detection Φ-OTDR and ROTDR system for acoustic and temperature sensing in 2018 [206]. Figure 18 gives the experimental setup of the hybrid sensing system. The Rayleigh, Raman Stokes, and Raman anti-Stokes components were separated by a WDM filter. Afterward, the Rayleigh scattering light was combined with the local oscillator light to reconstruct the acoustic field. Simultaneously, the Raman Stokes and anti-Stokes lights were detected by two APDs. A common data acquisition card was employed to record the Rayleigh and Raman signals together.

The hybrid Φ-OTDR/ROTDR can measure simultaneously in a single-mode fiber (SMF). However, the SpRS power is usually ~30 dB lower than the Rayleigh scattering light. To guarantee sufficient SpRS power, high pump power is needed, which may generate nonlinear effects and distort the Φ-OTDR trace [207]. Since the required pump powers for Φ-OTDR and ROTDR are incompatible in an SMF, a space-division multiplexed hybrid Φ-OTDR/ROTDR system using MCF was proposed and demonstrated by Zhao et al. in 2018 [208]. As illustrated in Figure 19, the laser source and the pump pulse were shared for Φ-OTDR and ROTDR, and the interrogations of Rayleigh and Raman scattering were carried out in different cores of the MCF. As a result, the Rayleigh and Raman signals were separated and detected independently. Moreover, it has been confirmed that the outer cores of the MCF were more sensitive to bending, which could be exploited for Φ-OTDR to exert an enhanced vibration.

### 3.3. Combination of Brillouin and Raman Scattering Lights

Since the Raman scattering is sensitive to temperature but not to strain, the combination of Brillouin and Raman scattering lights could help to address the cross-sensitivity issue in Brillouin scattering-based sensors. The intensity trace of the temperature-dependent Raman anti-Stokes light is normalized by the temperature-insensitive Raman Stokes trace, which has already been illustrated in Equation (17). As long as the Brillouin frequency shift is obtained simultaneously, the strain experienced by the fiber can be resolved with the assistance of the temperature trace *T*(*z*). The obtained strain trace can be expressed as:(20)$$\varepsilon (z)=\frac{\Delta \nu_{B}-C_{T}^{B}T(z)}{C_{\varepsilon }^{B}}.$$

The hybrid Brillouin and Raman sensor has been proven as an effective solution to discriminate distributed temperature and strain. In 2005, Alahbabi et al. presented a hybrid BOTDR/ROTDR sensing system for the first time [209]. The two measurands, temperature and strain, were successfully discriminated by combining the Brillouin and Raman scattering lights. Nevertheless, the Raman scattering light possesses rather low power, which leads to inaccuracy in simultaneous temperature/strain sensing. In 2013, Taki et al. presented a hybrid BOTDA/ROTDR sensing system based on cyclic pulse coding [210]. The use of a 511-bit cyclic simplex codeword brought a ~10 dB gain of SNR, allowing us to improve the accuracy of both Raman anti-Stokes and BFS measurements. A highly integrated solution was proposed, as illustrated in Figure 20. The coded pulses were shared for both BOTDA and ROTDR. A high extinction Raman filter was employed to separate the Raman Stokes, Raman anti-Stokes, and Rayleigh/Brillouin components into three different ports. Then, a narrowband FBG filtered out the unnecessary components, reserving the Brillouin Stokes light. Notably, since the Simplex coding bit sequences were quasi-periodic, the real-time decoding could be achieved in less than one fiber transit time, allowing accurate long-distance sensing with a sub-second measurement time [211].

However, an extremely high pump power (several Watts) is required to increase the intensity of SpRS. The nonlinear tolerance of ROTDR is much higher since the Raman Stokes and anti-Stokes signals have a broader spectral width. However, the Brillouin scattering-based sensors will suffer from modulation instability (MI) and SBS effects. The Brillouin scattering trace could undergo dramatic distortion resulitng from these nonlinear effects [207].

To address the incompatibility of the pump power, Zhao et al. utilized the SDM in a hybrid BOTDR/ROTDR sensing system based on multi-core fiber [212]. The interrogation of the BOTDR and ROTDR sub-systems was operated at different cores of the multi-core fiber. As exhibited in Figure 21, the central core was chosen for BOTDR to avoid the bending-imposed BFS cross-sensitivity, while one of the outer cores implemented ROTDR measurement. Therefore, the quantitative discrimination of temperature and strain was guaranteed. Benefitting from the SDM, the power of pump pulses at BOTDR and ROTDR branches could be separately controlled, which were set to 20 dBm and 33 dBm, respectively.

### 3.4. Others

Apparently, the hybrid DOFSs mentioned above are all based on the independent detection of different scattering lights. In addition to the direct combination methods, an alternative approach is to gather multiple scattering information demodulated from a single scattering light.

In 2013, Wang et al. proposed a distributed fiber strain and vibration sensor integrating BOTDR and polarization optical time domain reflectometry (POTDR) [213]. Figure 22 depicts a schematic diagram of the hybrid BOTDR/POTDR system, which resembled a conventional heterodyne BOTDR configuration, whereas the principle of POTDR was exploited by extracting the state of polarization (SOP) of Brillouin scattering light. A polarization switch (PSW) instead of a PS was used to manipulate the polarization state of the reference light. When the PSW was turned to a certain state, the instant polarization information was demodulated from the Brillouin trace. A subsequent fast Fourier transform (FFT) was performed to get the distributed frequency spectrum of the polarization signal, revealing the location and frequency of the vibration. When the PSW had accomplished a periodic switchover (both horizontal and vertical SOPs), the detected Brillouin scattering light with two orthogonal SOPs could suppress the polarization fading noise, allowing for the correct measurement of temperature and strain.

Later in 2015, Wang et al. presented a Rayleigh scattering-based DOFS capable of measuring strain and vibration simultaneously [214]. Figure 23 depicts the experimental setup of this sensing system. The combination of an EOM and a microwave source provided frequency-sweeping pulses for Φ-OTDR operating in the cross-correlation measuring mode. To conduct a simultaneous multi-parameter measurement, the pump pulse was initially fixed, and the real-time Rayleigh traces were obtained to measure dynamic vibration. Afterward, the frequency-sweeping pulses were generated, along with the multiple Rayleigh traces carrying different Rayleigh frequency shifts. Whenever the scanning procedure was accomplished, the cross-correlation peak could be acquired to determine the distributed temperature/strain, as illustrated in Equations (8) and (9), and the measurement of dynamic vibration was also guaranteed.

Recently, Fan et al. proposed a simplified single-end hybrid Rayleigh and Brillouin sensing system [215]. Unlike the conventional scheme, demodulation was only conducted towards the Rayleigh scattering light, which contained the Brillouin scattering information as well. As depicted in Figure 24, the DFB laser performed injection locking to compensate for the insertion loss of the intensity modulator (IM) and the optical bandpass filter (OBPF), and to stabilize the power of the output light. Another difference from the conventional BOTDA configuration is that the pulse pair was comprised of two segments, and the former pulse initially generated the continuous RBS light, which also acted as the Brillouin probe. Once the frequency difference between the former and latter pulses came near the BFS, the latter pulse was regarded as the Brillouin pump pulse to amplify the probe due to the SBS effect. The continuous RBS light, which carried the Rayleigh scattering information, was also used to demodulate the BFS. As a consequence, only one type of signal was detected to obtain the vibration measured by Φ-OTDR and the temperature/strain measured by BOTDA. Benefitting from its well-designed structure, the BOTDA sub-system could be implemented from the single end of the fiber, and the requirement of the receiver was reduced.

To sum up, hybrid DOFSs based on the combination of Rayleigh, Brillouin, and Raman scattering lights have a number of sensing abilities:(1)Measurements of multiple parameters, including temperature, strain, vibration, and acoustic wave;(2)Discriminating temperature and strain for simultaneous measurements;(3)Improving the performance of single-parameter (e.g., temperature, strain) measurement based on the simultaneous demodulation of Rayleigh and Brillouin signals.

In order to realize the integration of various DOFS systems, efforts have been made in consideration of:(1)The direct separation of different scattering lights with FBG or WDM. The Raman frequency shift is large enough to take advantage of the WDM. However, the Brillouin frequency shift is much smaller, so it is rather necessary to use the FBG with ultranarrow bandwidth to separate Brillouin and Rayleigh scattering lights;(2)Well-designed multiplexing methods incorporating time–division multiplexing, wavelength–division multiplexing, space–division multiplexing, frequency–division multiplexing, or polarization–division multiplexing.

This section reviews the recent advances of the hybrid DOFS technology. To integrate various DOFS systems with different scattering mechanisms, major works focus on separating and detecting Rayleigh, Brillouin, and Raman scattering lights. Different combinations of scattering lights require specific methods due to their distinct principles.

## 4. Multi-Parameter Measurements

While implementing in-field tests, the greater the amount of parameters that are measured, the better the recognition of the fault events. Benefitted by the hybrid DOFSs presented in Section 3, multiple parameters, including temperature, strain, vibration, and loss, could be measured simultaneously, allowing for extended applications in engineering.

### 4.1. Temperature/Strain and Vibration

Basically, simultaneous temperature/strain and vibration sensing can be realized by a hybrid Rayleigh and Brillouin sensor.

As demonstrated by Coscetta et al. [192], both the BFS and the intensity information of the Rayleigh signal were resolved. Figure 25a shows the BFS profiles acquired from the BOTDA data. Results under different temperature conditions are marked in distinct colors. The temperature-induced BFS variation was clearly observed at the fiber end, indicating a 2 m spatial resolution. The corresponding temperature revolution is also given in Figure 25b. Notably, the BOTDA setup was adjusted to a slope-assisted BOTDA configuration, which could conduct dynamic strain measurements. For comparison, the dynamic strain and the vibration tests were synchronously implemented through BOTDA and Φ-OTDR, respectively. As shown in Figure 25c,d, the Φ-OTDR measurement was much cleaner than the BOTDA measurement when imposing the same vibration (150 Hz sine wave with the peak-to-peak value of 1.5 V). A possible explanation was that the strain sensitivities of BOTDA and Φ-OTDR were ~0.25%/με and 10%/nε, respectively. Thus, BOTDA could be used to estimate the absolute amount of strain, while the vibration spectrum was determined by Φ-OTDR with more fidelity. The combination of Φ-OTDR and slope-assisted BOTDA could be a promising method to analyze and reconstruct the vibration.

Simultaneous temperature/strain and vibration measurements were also carried out using the experimental setup in Figure 10 [145]. Figure 26a illustrates the arrangement of the FUT with a total length of 59.9 km, involving a fiber section with a 10 m length wrapped on a piezoelectric transducer (PZT) tube and another fiber loop with a 20 m length immersed in a water bath kettle. Both events were located near the end of FUT. Figure 26b shows the demodulated phase of the RBS signal when an 800 Hz triangular vibration signal was exerted on the PZT; it is evident that the dynamic vibration information was restored correctly. At the same time, the temperature of the water bath kettle rose from 5 °C to 50 °C, which could be observed from the corresponding measurement results in Figure 26c. The measurement uncertainty of BFS was 0.381 MHz, and the achieved dynamic strain resolution was 1.235 nε/√Hz when the spatial resolution of the hybrid sensing system was 20 m.

Besides this, the hybrid Φ-OTDR/BOTDA sensing system reported by Fu et al. [194] implemented multi-parameter measurements over a hundred-kilometer length scale. The BFS profiles are illustrated in Figure 27a, which manifested a BFS variation near the fiber end, corresponding to an 18.2 °C temperature difference in a 35 m fiber piece. The enlarged view of the heated section in Figure 27b validated that the spatial resolution was 8.2 m for temperature measurement. It should be noted that it can also measure the strain since both were derived from the BFS. The vibration measurement was tested by applying an external perturbation near the fiber end. Figure 27c gives the demodulated intensity distribution of the Rayleigh signal, where the perturbation was well recognized. According to Figure 27d, the spatial resolution for vibration measurement was 28.8 m, while a sensing range of 150.62 km was achieved with the assistance of multiple distributed amplification.

### 4.2. Temperature and Vibration

As discussed in the previous section, simultaneous measurements of temperature and vibration are required in numerous application scenarios, especially in the oil and gas industry and geophysical engineering.

The hybrid Φ-OTDR/ROTDR sensing system in Figure 19, which took advantage of the multi-core fiber, has accomplished simultaneous distributed intrusion detection and temperature monitoring [208]. To emulate the intrusion events, a manual tapping was imposed on two segments of the fiber. The experimental results are shown in Figure 28. Figure 28a depicts the superposed 885 consecutive signals based on the differential operation between raw intensity traces and a reference trace, revealing the disturbance locations corresponding to the two intensity peaks. In the meantime, the temperature measurement was carried out by ROTDR in the central core of the multi-core fiber. Different temperatures (50 °C, 60 °C, 75 °C) were applied on the 60 m-long fiber at the fiber end, and Figure 28b depicts the results with the assistance of the wavelet transform denoising (WTD) method. The temperature distribution was precisely resolved, especially when the WTD was used. Notably, the overall performance of the multi-parameter measurements was more impressive with the assistance of space-division multiplexing, and the use of a single MCF avoids the access of multiple SMFs, leading to a reduction in the cost associated with embedding or attaching the sensors. It also results in simpler and more robust sensor design and thus an increased range of structural applications.

It is noteworthy that the multi-core fiber is not always suitable for and compatible with engineering, especially when the fiber type has already been determined. The simultaneous temperature and vibration measurements have also been conducted in SMF by Zhang et al. through a hybrid Φ-OTDR/ROTDR sensing system [206]. The method and the experimental setup have already been described in Section 3.2. The detected Raman Stokes and anti-Stokes signals are shown in Figure 29a. As the fiber end was heated from 35 °C to 55 °C with the step of 5 °C, using the temperature control chamber, temperature distribution could be obtained by combining the Raman Stokes and anti-Stokes signals. Figure 29b gives an enlarged view of the heated section, where the spatial resolution of 10 m was confirmed. The vibration test was first carried out by knocking the chamber periodically, resulting in a periodic intensity change as illustrated in Figure 30a. Simultaneously, a sinusoidal vibration was applied on a cylindrical PZT wrapped with a 10 m long fiber piece. Since the Φ-OTDR was operating at the heterodyne detection configuration, vibration with different amplitudes could be restored based on the phase demodulation, as shown in Figure 30b, indicating competence in measuring vibration-induced dynamic strain. Moreover, vibration with different frequencies was also analyzed in Figure 30c, fully proving the ability of dynamic strain sensing.

### 4.3. Simultaneous Temperature and Strain

Even though the Brillouin scattering-based DOFSs are able to resolve temperature and strain by BFS, they cannot be regarded as multi-parameter measurements since the BFS is fundamentally linear to both temperature and strain. Therefore, cross-sensitivity is a real drawback during monitoring. In order to discriminate the temperature and strain, plenty of methods have been proposed over the last few decades [180,216,217,218,219,220,221]. It has been proven that the Brillouin scattering light assisted by Rayleigh or Raman scattering light can be an effective solution to achieve the discrimination; meanwhile, the specialty fibers are not required.

As mentioned in Section 3.1, the hybrid BOTDR/COTDR, called BOTDR distributed strain and temperature sensing (DSTS) by Clément et al., utilized the Landau–Plazek ratio to handle the problem of cross-sensitivity [198]. Two cylinders made from different materials were designed to apply temperature and strain simultaneously. Figure 31a shows the fiber deployment. Both cylinders (wrapped in a 125 m long fiber) and the 200 m unstrained optical fiber were placed in a climatic chamber to vary the temperature. The cylinders made of aluminum and steel showed different thermal expansion coefficients, leading to a distinct strain on the fiber. It could be observed from Figure 31b that the first cylinder expanded more than the second under the same temperature (red curve versus purple curve between two areas), and the temperature measurement (blue curve versus yellow curve) was clearly distinguished. As long as the simultaneous measurement of temperature and strain was realized, further in-field tests were implemented. An in-well measurement was conducted to validate the performance, which is illustrated in Figure 32a. The interrogators (BOTDR DSTS) were connected to one end of a cable, which was coiled and inserted vertically into the well (2 km depth). The cable would undergo several forces in a static regime, such as the weight of the tool or the self-weight of the cable, the frictional force due to the possible rough walls of the well, Archimedean thrust (well generally filled with different fluids), and thermal expansion or hydrostatic pressure force, which is difficult to predict and quantify [222]. For comparison, temperature measurement was also carried out by a standard ROTDR and a standalone BOTDR. From Figure 32b, we can see that the temperature distributions measured from ROTDR (red curve) and the proposed DSTS (blue curve) were almost identical, manifesting the good discrimination of temperature from strain. However, the temperature measurement from the standalone BOTDR (yellow curve) brought about obvious errors due to the strain accumulation. On the other hand, the actual strain distribution (purple curve) revealed a significant strain variation located at the bottom of the well (~4.8 km).

Since the Rayleigh and Brillouin frequency shifts have distinct dependencies on temperature and strain, the hybrid Rayleigh and Brillouin system is endowed with another method to discriminate these two measurands. Murray et al. conducted a series of measurements based on the experimental setup introduced in Figure 15 [172]. The test fiber was 460 m long, and a single section of fiber (8 m length) was wrapped around a PZT. A sinusoidal signal with 50 Hz frequency and ~600 peak-to-peak nε drove the PZT. Meanwhile, the PZT was placed on a temperature stage to impart a simultaneous temperature variation. Benefitting from the discrimination method, vibration-induced strain and temperature located at the same fiber section could be observed. Figure 33a,b exhibited the recovered dynamic temperature and strain, respectively. The fitted line in Figure 33a indicates a gradual temperature increase, while the demodulated strain curve in Figure 33b shows the elimination of a bias introduced by temperature. In other words, the cross-sensitivity problem has been successfully solved. The amplitude spectral density (ASD) curves based on the recovered temperature and strain were also obtained, as shown in Figure 33c,d. It is evident that the levels of noise floor could be ued to calculate the temperature and strain measurement sensitivities, which were 0.54 m°C/√Hz and 4.5 nε/√Hz, respectively.

Another measurement was carried out by Taki et al. [210] based on the hybrid Brillouin/Raman sensing system in Figure 20. The FUT was arranged with 15 m of fiber segment heated to 60 °C at the fiber end (~10 km total length), and the rest of the fiber was at room temperature (32 °C during the test). To discriminate the temperature profile, the ROTDR sub-system performed the independent temperature measurement. Figure 34a depicts the distribution curve; a ~28 °C temperature step can be observed at the corresponding heating area. The worst calculated temperature resolution was ~3.4 °C with 3 million average times. Then, the strain could be subsequently measured by the BOTDA sub-system using Equation (11), leading to a temperature-independent estimation of strain. As depicted in Figure 34b, the worst strain resolution was ~80 με. Notably, a meter-scale spatial resolution was achieved over kilometers of sensing range since cyclic pulse coding was employed to enhance the SNR. In addition, Taki et al. conducted an improved experiment in 2014 [223], in which a linear translation stage with an external thermoelectric jacket was arranged to apply temperature and strain variations to the same spool of fiber. The capacity to discriminate BFS changes on the same fiber section was validated. It is clear that the combination of temperature and strain would offer more valuable information for structure monitoring.

### 4.4. Comprehensive Temperature Measurement

It should be noted that although Brillouin sensors as well as the frequency-scanning Rayleigh sensors can measure temperature, the resolution and sensing range are quite different. For instance, the typical measurement accuracy of BFS in Brillouin sensors is within the megahertz scale, corresponding to a temperature resolution of approximately 1 °C [200]. Meanwhile, the measurement range is considerable. On the other hand, the frequency-scanning Rayleigh sensors have been proven at a resolution of about 0.01 °C or even higher [173], yet the measurement range is restricted within the magnitude of 1 °C. Therefore, the combination of these two kinds of sensors has the potential to realize a large dynamic range as well as an ultrahigh measurement resolution.

As previously presented, the hybrid Φ-OTDR/BOTDA sensing system demonstrated by Dang et al. [200] achieved a comprehensive temperature measurement. Experiments were conducted on a 1.565 km multi-core fiber whose attenuation coefficient was ~0.25 dB/km. Two short segments (section A and section B, length of 37 m) at the far end were separately immersed in two water baths to apply different temperatures. Figure 35a,b give the experimental results measured by BOTDA and Φ-OTDR, respectively. BOTDR cannot distinguish the natural cooling-down process in section A, whereas the large temperature variation (30 °C increment with a step of 10 °C) at section B was well-recognized. On the other hand, the slight temperature difference of 0.1 °C was identified by Φ-OTDR. However, an error occurred when large temperature variation was applied at section B, since the frequency scanning range of Φ-OTDR was too narrow to compensate for the temperature-induced refractive index change. The temperature accuracy of BOTDA/Φ-OTDR was also estimated by calculating the standard deviation (STD) of BFS distribution or cross-correlation peak frequency shift, as illustrated in Figure 35c,d. Compared with the ~0.25 °C uncertainty of BOTDA, Φ-OTDR achieved an ultrahigh temperature resolution of about 0.001 °C. The above experiments have demonstrated the capability of the hybrid sensing system to measure temperature with a large dynamic range and high resolution simultaneously. It should be pointed out that other parameters can also be measured besides temperature. Considering the significance of comprehensive temperature sensing, we regard it as an application of multi-parameter measurement.

### 4.5. Comprehensive Strain Measurement

Apart from temperature, strain is another measurand of Brillouin sensors and frequency-scanning Rayleigh sensors. Their relations to the resolution and sensing range are similar to the conditions when measuring temperature. Wang et al. [199] demonstrated a hybrid BOTDA/Φ-OTDR sensing system by combining the Rayleigh and Brillouin scattering lights (presented in Figure 7), allowing for simultaneous measurements of absolute strain and relative strain. The vibration was applied to the FUT by changing the motorized positioning system between two positions periodically. The relative displacement was 1 μm, corresponding to a 500 nε strain. In addition, a fixed large strain, which represented the absolute strain, was initially applied. Then, two groups of experiments were conducted with the same vibration under different absolute strains. The measurement of the absolute strain by BOTDA is depicted in Figure 36a, where different strains in two groups are observed. However, the BOTDA cannot sense the sub-micro strain due to the relatively low strain resolution. Meanwhile, the relative strain change could be demodulated through the frequency-scanning Φ-OTDR. As exhibited in Figure 36b,c, a 9.9 Hz vibration with a peak-to-peak value of 500 nε was obtained. Thus, the relative strain and absolute strain could be combined to demonstrate the exact strain evolution, which is displayed by the right y-coordinates of Figure 36b,c. The measurement accuracies of the relative strain and the absolute strain were estimated to be 6.8 nε and 5.4 με, respectively. As a result, the BOTDA and Φ-OTDR are complementary schemes rather than substitutes for each other, such that further cooperation can extend the functions and reach into new application areas [199].

This section introduces the multi-parameter measurement performance, which is based on the hybrid DOFS technology described in Section 3. By combining different hybrid DOFS systems, the measurable parameters and their features are also made distinct. Typical results are summarized in Table 1 with a list of relevant references.

## 5. Discussions

### 5.1. Challenges

(a)Difficulty in separating backscattering lights

As mentioned in Section 3, separating different backscattering lights is a key process to realize the hybrid DOFS system. Since the Raman scattering light has a frequency shift of several nanometers, the DWDM can easily extract the Raman scattering light. However, the frequency difference between the Rayleigh and Brillouin scattering lights is much narrower in the wavelength domain. The FBG filter and the Mach–Zehnder interferometer [224,225,226] are generally used to separate the two frequency components. In addition, it is also practicable to process the scattering signal in the frequency domain. A photodetector with chosen bandwidth can act as the electrical filter [198]. Nevertheless, the performance of the optical signal processing is limited by the state-of-the-art manufacturing technique. The bandwidth of the FBG filter must be narrower than the BFS, and the reflectivity and the rejection level should be high enough to prevent the crosstalk between Brillouin and Rayleigh scattering lights. As to the interferometer, it is too sensitive to external fluctuation, which is an obstacle in industrial applications. A possible solution is to integrate the silicon-based waveguide with the interferometer structure to maintain stability, but the insertion loss is still non-ignorable. With the rapid growth of optoelectronic device fabrication, more effective solutions could be developed to separate the different sensing signals.

(b)Incompatibility in pump power requirements

Another notable problem is the distinct requirement of pump power among different sensing systems. As previously mentioned, the Raman scattering-based DOFS normally needs a rather high pump power to generate considerable SpRS light. In the meantime, it would induce nonlinear effects in the Rayleigh and Brillouin scattering-based DOFSs, leading to significant errors. The multi-core fiber is an appropriate method to integrate different sensing systems with discrete pump power paths. However, there is no industrial standard for multi-core fiber manufacturing so far. As a result, multi-core fiber is not a common choice in various application scenarios, especially when laying new fiber is unacceptable. The most promising method may be the direct modulation of the pump pulses combining time–division multiplexing (TDM) or frequency–division multiplexing (FDM) techniques to evade the nonlinear effects.

### 5.2. Prospects

(a)Integration with frequency domain DOFS

Since the hybrid sensing systems mainly consist of DOFSs based on the OTDR technique, the spatial resolution is restricted by the width of the pump pulse. On the other hand, the OFDR technique can realize higher spatial resolution and SNR, but the sensing distance is limited. In recent years, a time-gated mode of digital optical frequency domain reflectometry (TGD-OFDR) [25,227,228] has been developed. It possesses the advantages of both OTDR and OFDR. The configuration of TGD-OFDR is similar to that of COTDR, except that the pulse is linearly frequency modulated. As a result, the frequency sweeping range determines the spatial resolution, which is as high as that in a conventional OFDR setup. At the same time, it achieves an improved sensing distance (up to 100 km) with the modulated pulse [227]. Considering TGD-OFDR has a similar configuration to COTDR, it represents a potential method to be adopted in a hybrid DOFS system and can largely improve the performance.

(b)Extended parameters sensing

Limited by the intrinsic sensitivity to external parameters of conventional silica optical fibers, current DOFS systems are mainly employed to measure loss, temperature, strain, and vibration. Applications in the oil and gas industry and environmental monitoring call for distributed chemical, gas, and humidity sensing. In recent years, the discrete BOTDA [229] and chirped-pulse Φ-OTDR [230] have demonstrated their competence in detecting hydrogen with standard telecom-grade optical fiber. Moreover, specialty sensing fibers have hugely extended the potential measurands due to their unique properties, for instance, the multicore fibers and ring-core fibers for distributed curvature and bending sensing [231,232], the photonic crystal fibers for distributed hydrostatic pressure sensing [233,234,235], and the polymer optical fibers for distributed humidity sensing [236,237] and distributed radiation sensing [238,239]. The hybrid DOFS technology, which combines various standalone sensors to measure more abundant parameters, is a promising tool to explore further applications that will meet the requirements in many different scenarios. In addition, since the Rayleigh DOFS and the Brillouin DOFS share common measurable parameters (e.g., humidity [236,237]), it is possible to merge the data from both sides in a hybrid DOFS system incorporating specialty fibers, thus enabling an improved measurement performance.

In recent years, the over-spraying of pesticides onto crops has escalated the pesticide contamination of food products and water bodies, as well as disturbing ecological and environmental systems [240]. The optical-sensing strategy offers high sensitivity and selectivity, allowing for the rapid detection of pesticides and heavy metal ions [241,242]. It is possible to develop specialty sensing fibers or to manufacture optical fibers with microstructures and carbon-coating, which is employed in a hybrid DOFS measurement scenario. As a result, not only can a variety of chemicals be simultaneously sensed along the entire arranged fiber, but the hybrid DOFS approach is also much more cost-effective.

(c)Potential to multiply overall performance with optimized methods

There has been an ongoing focus on the performance improvements of DOFS systems, in terms of spatial resolution, sensing distance, SNR-related measurement accuracy, measuring speed, sensitivity, and so on. Various methods, involving the modulation of the pump pulse, demodulation, and data-processing algorithms, have been proposed and utilized to enhance the performance of an individual sensing system. However, since different DOFSs are operated based on different sensing principles, their configurations and components have disparities. The incompatibility problem may occur when using a certain method to improve one of the sub-systems. Thus, a well-designed method for the enhancements of two or more sensing systems is crucial, and it multiplies the performance and efficiency as well.

(d)Merging valuable information for further applications

There is no doubt that the multi-parameter measurement has multiplied the dimension of the acquired data sets. As a result, merging the multidimensional data could be a promising method to improve the performance when sensing the same measurand. Lately, Ba et al. exploited the potential of the hybrid Φ-OTDR/BOTDA sensing system [201]. The strain traces Δ*ε_R_* and Δ*ε_B_*, which were demodulated from Φ-OTDR and BOTDA, respectively, served as input parameters of an adaptive signal corrector (ASC). The ASC output *y* was generated after the filtering process. To reach the optimized result, the parameters of the filter were iterated and adjusted adaptively according to the difference between Δ*ε_B_* and *y*. As the measurement time increased, an accumulated measurement error, which was suggested as a common appearance in frequency-scanning Φ-OTDR [243], could be observed from the Rayleigh trace. Such a problem was well addressed with the assistance of ASC, since the experimental results have shown successful fast pressure tracking from 20 MPa to 0.29 MPa without a measurement error. Meanwhile, the calculated STD of pressure distribution indicated a pressure measurement resolution of 0.14 kPa.

Once the multi-parameter data sets are obtained, it is much more advantageous to evaluate the condition of the monitored objects when combined with machine learning algorithms and classifiers [19,140,244,245,246,247,248]. During the monitoring, different types of events emerge simultaneously, in which there also exist irrelevant interference events. The pattern classification algorithm can be used to distinguish various events and recognize the targeted events. Normally, there are two methods to reach recognition, including feature value extraction [140] and image recognition [246]. However, they are mainly based on the data from single parameter measurements, indicating the lack of feature values or the inaccuracy of recognition. Wang et al. [141] proposed to monitor the leakage of oil pipelines by using both temperature and vibration information, and studied the effect in a simulated experiment. They classified the states of pipelines into three types, including the normal state, the leakage state, and the interference state. Then, 11 kinds of feature values were chosen to recognize the pipeline states. Benefitting from the multi-parameter measurement, the comprehensive recognition rate reached 98.57%, which is a great improvement compared with the recognition under individual parameters. The combination of multi-parameter measurements and machine learning algorithms can exploit further applications in modern industrial practice.

In this section, the challenges and prospects of the hybrid DOFS technology are elaborately discussed. It is challenging to directly separate different scattering lights, as well as to address the existing incompatibility issue of different DOFS systems. Nevertheless, novel techniques, such as the multiplexing method, advanced manufacturing technology, and the use of specialty sensing fiber, could greatly extend the range of applications. The hybrid DOFS technology can not only continually improve the multi-parameter measurements performance, but the merging of information related to any measurable parameter will also come to show greater potential.

## 6. Conclusions

In this review, we summarize and discuss recent advances in the study of hybrid distributed optical fiber sensors for multi-parameter measurements. This work is concentrated on the principle, integrating and optimizing methods, as well as multi-parameter measurement performance regarding the hybrid DOFS. The hybrid DOFS technology is mainly based on the combination of the Rayleigh, Brillouin, and Raman scattering mechanisms, presenting competence in measuring multiple parameters, including temperature, strain, and vibration. With the progress of employing multiple novel methods, the performance of the hybrid DOFS is further enhanced. It is believed that distributed multi-parameter sensing could multiply the valuable information derived from the measurands, contributing to a much more efficient and accurate recognition of the potential events. Compared to the single DOFS system, the utilization of multiple parameters based on the hybrid DOFS is highly beneficial to the monitoring of large facilities, reducing the emergence of misinformation and false alarms. The hybrid DOFS technology has manifested great competence in the various aspects of industrial areas, which are expected to be exploited in further applications.

## Figures and Tables

**Figure 1 sensors-23-07116-f001:**
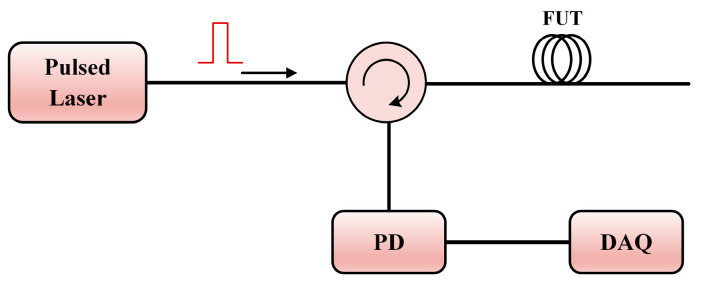
The basic configuration of OTDR. DAQ: data-acquisition card.

**Figure 2 sensors-23-07116-f002:**
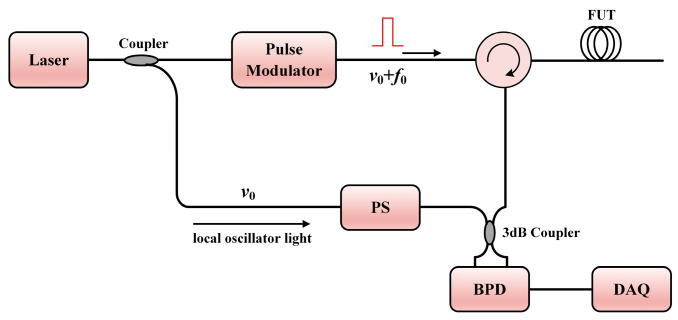
The basic configuration of COTDR.

**Figure 3 sensors-23-07116-f003:**
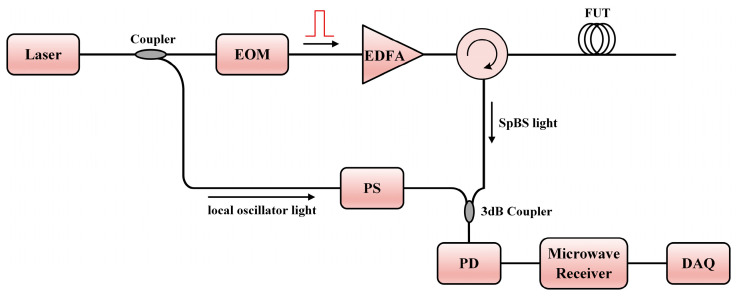
The typical configuration of BOTDR with coherent detection. EOM: electro-optic modulator; EDFA: erbium-doped optical fiber amplifier.

**Figure 4 sensors-23-07116-f004:**
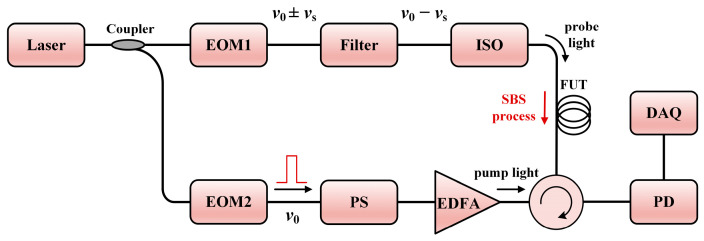
The typical configuration of BOTDA. ISO: isolator.

**Figure 5 sensors-23-07116-f005:**
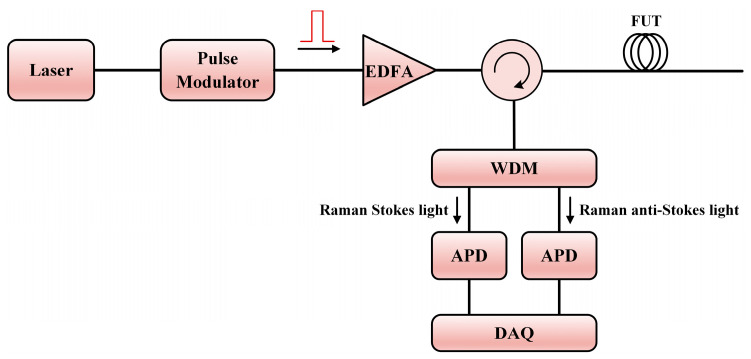
The typical configuration of ROTDR. WDM: wavelength-division multiplexer; APD: avalanche photodiode.

**Figure 6 sensors-23-07116-f006:**
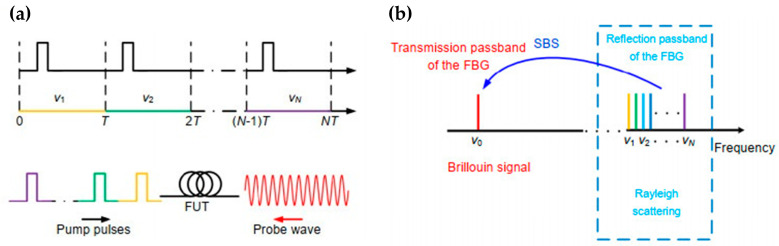
(**a**) Diagram of the frequency-scanning pump pulses and the probe wave. (**b**) Frequency relationship between the Rayleigh and the Brillouin scattering signals (reprinted from [199], under the terms and conditions of the Creative Commons Attribution License).

**Figure 7 sensors-23-07116-f007:**
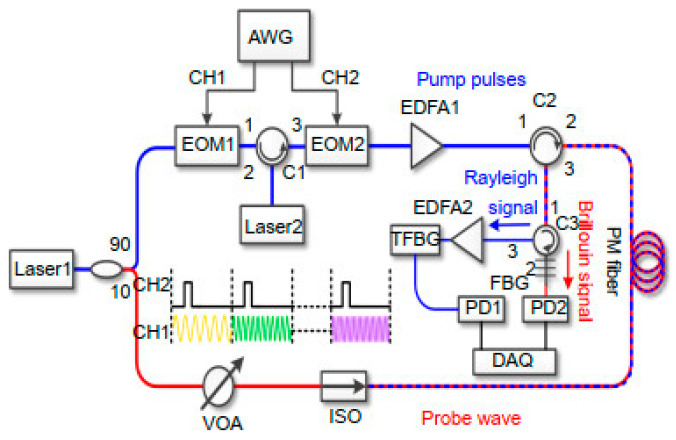
Experimental setup of the hybrid Φ-OTDR/BOTDA system (reprinted from [199], under the terms and conditions of the Creative Commons Attribution License).

**Figure 8 sensors-23-07116-f008:**
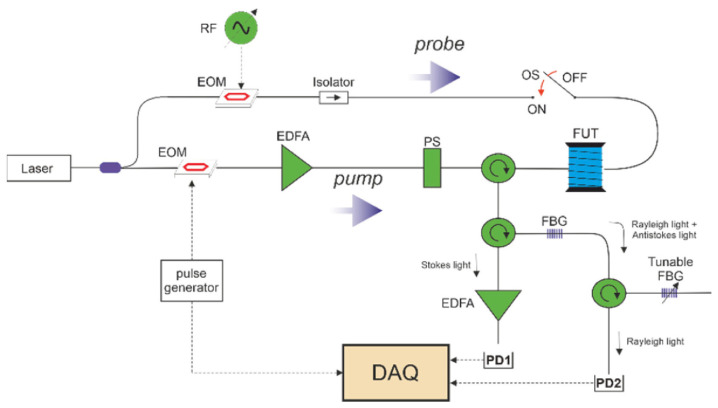
Experimental setup of the hybrid Φ-OTDR/BOTDA system (reprinted from [192], under the Open Access Publishing Agreement from © 2021 Optical Society of America).

**Figure 9 sensors-23-07116-f009:**
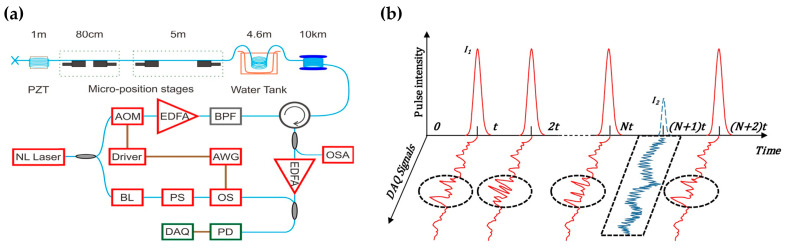
(**a**) Experimental setup of the hybrid Φ-OTDR/BOTDR system. (**b**) Modulated pulse sequences and the corresponding acquired signals (reprinted from [193], under the Open Access Publishing Agreement from © 2016 Optical Society of America).

**Figure 10 sensors-23-07116-f010:**
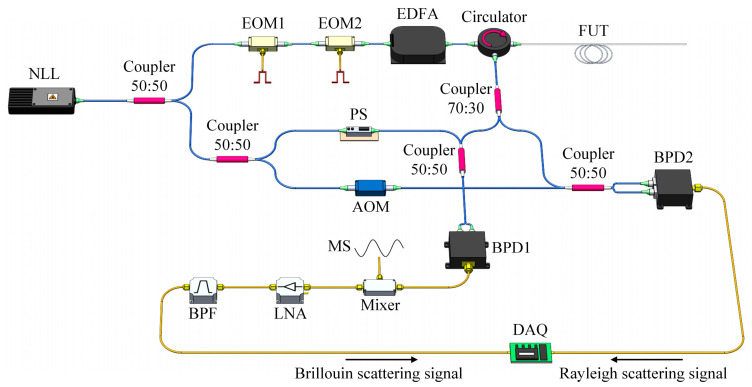
Experimental setup of the hybrid Φ-OTDR/BOTDR system (reprinted from [145], under the Open Access Publishing Agreement from © 2022 Optica Publishing Group).

**Figure 11 sensors-23-07116-f011:**
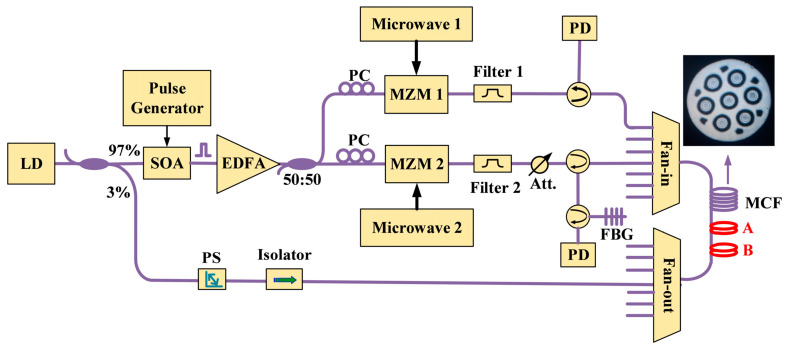
Experimental setup of the hybrid Φ-OTDR/BOTDA system through space-division multiplexing (SDM) based on the multi-core fiber (MCF) (reprinted from [200], under the Open Access Publishing Agreement from © 2017 Optical Society of America).

**Figure 12 sensors-23-07116-f012:**
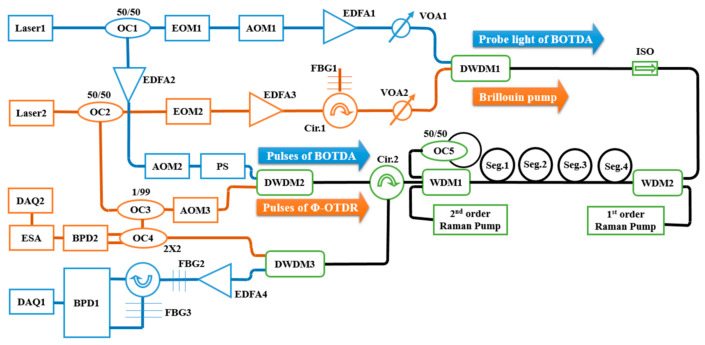
Experimental setup of the hybrid Φ-OTDR/BOTDA system combining multiplexing and distributed amplification techniques (reprinted from [194], under the terms and conditions of the Creative Commons Attribution License).

**Figure 13 sensors-23-07116-f013:**
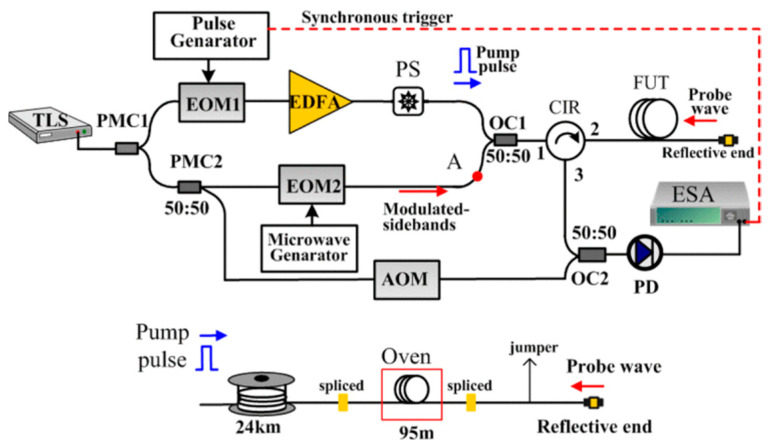
Experimental setup of the hybrid single-end-access BOTDA and COTDR system (reprinted from [195] with permission, © 2013 IEEE).

**Figure 14 sensors-23-07116-f014:**
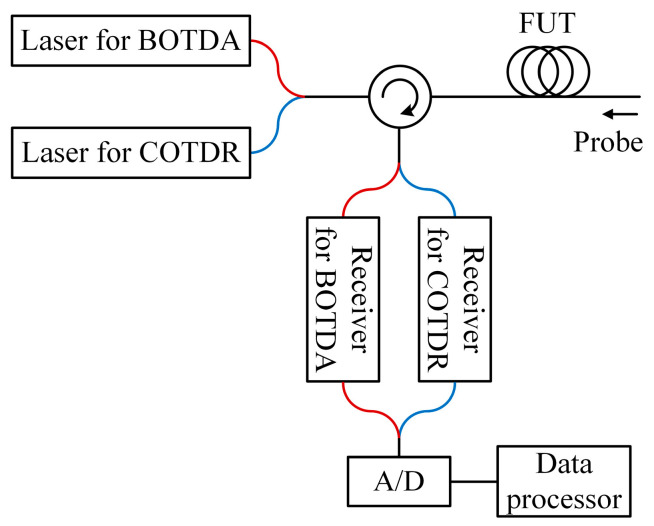
Experimental setup of the hybrid BOTDA/COTDR system (adapted from [197]).

**Figure 15 sensors-23-07116-f015:**
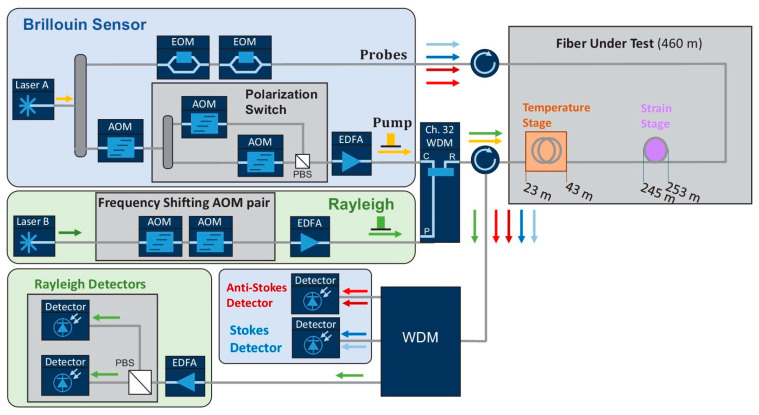
Experimental setup of the hybrid Rayleigh and Brillouin system (reprinted from [172], under the Open Access Publishing Agreement from © 2023 Optica Publishing Group).

**Figure 16 sensors-23-07116-f016:**
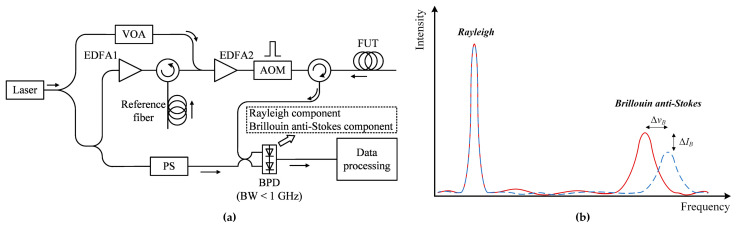
(**a**) Experimental setup of the hybrid BOTDR/COTDR system; (**b**) schematic representation of the detecting signal spectrum according to the temperature or strain change (adapted from [198]).

**Figure 17 sensors-23-07116-f017:**
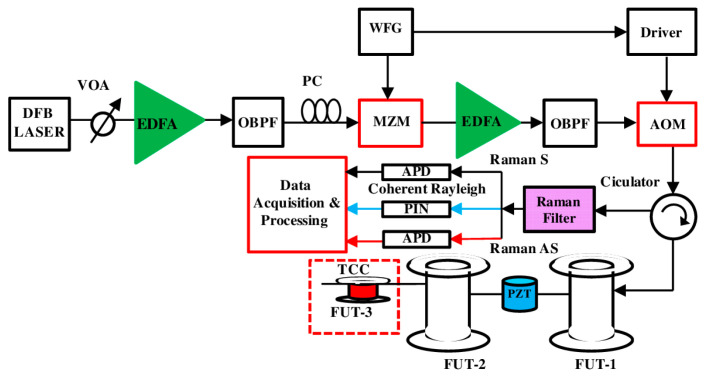
Experimental setup of the hybrid Φ-OTDR/ROTDR system using a commercial off-the-shelf DFB laser and direct detection (reprinted from [205] with permission, © 2016 Optical Society of America).

**Figure 18 sensors-23-07116-f018:**
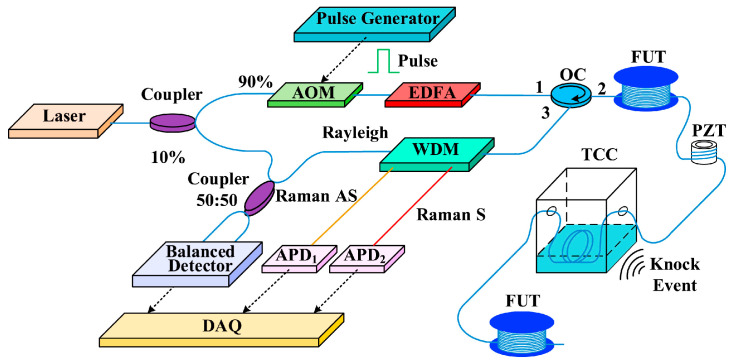
Experimental setup of the hybrid Φ-OTDR/ROTDR system (reprinted from [206] with permission, © 2018 Elsevier).

**Figure 19 sensors-23-07116-f019:**
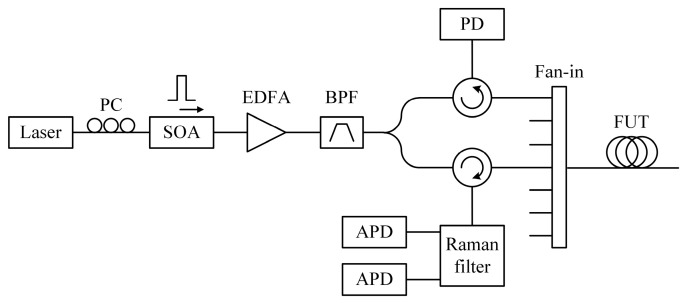
Experimental setup of the hybrid Φ-OTDR/ROTDR system based on multi-core fiber (adapted from [208]).

**Figure 20 sensors-23-07116-f020:**
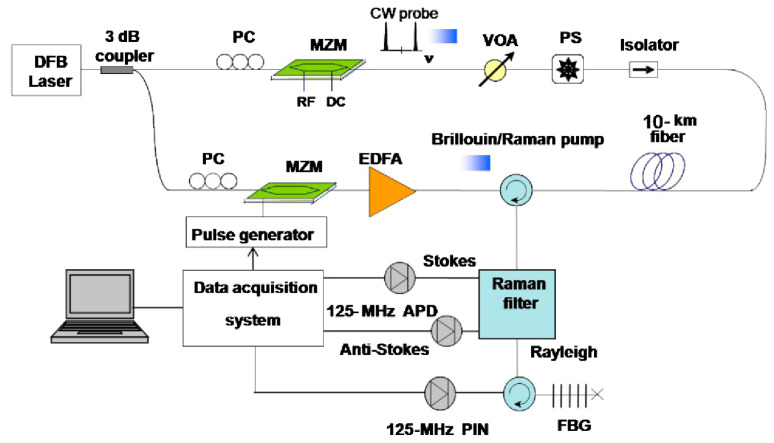
Experimental setup of the hybrid BOTDA/ROTDR system using cyclic pulse coding (reprinted from [210] with permission, © 2013 Optical Society of America).

**Figure 21 sensors-23-07116-f021:**
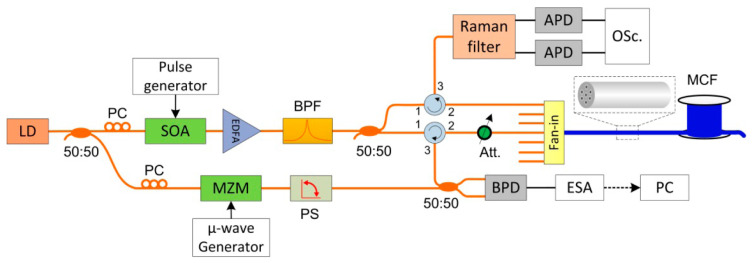
Experimental setup of the hybrid BOTDR/ROTDR system based on multi-core fiber (reprinted from [212], under the Open Access Publishing Agreement from © 2016 Optical Society of America).

**Figure 22 sensors-23-07116-f022:**
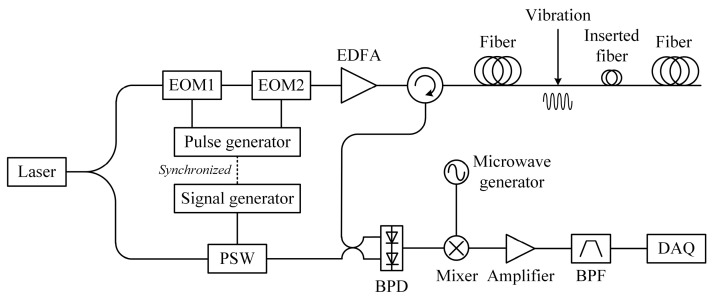
Experimental setup of the hybrid BOTDR/POTDR system (adapted from [213]).

**Figure 23 sensors-23-07116-f023:**
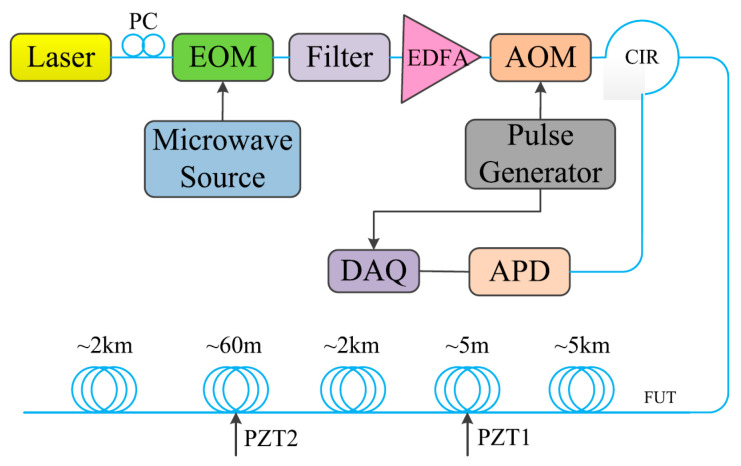
Experimental setup of the frequency-scanning Φ-OTDR (reprinted from [214] with permission, © 2015 IEEE).

**Figure 24 sensors-23-07116-f024:**
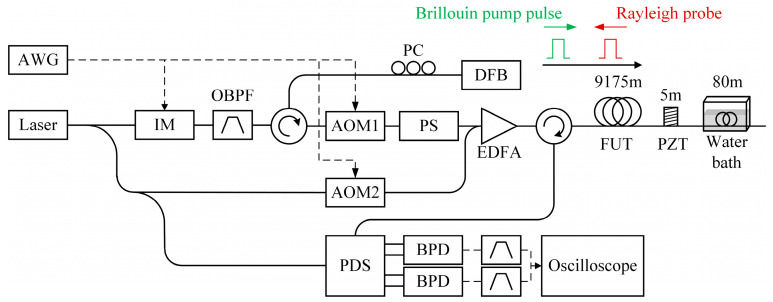
Experimental setup of the single-end hybrid Φ-OTDR/BOTDA system (adapted from [215]).

**Figure 25 sensors-23-07116-f025:**
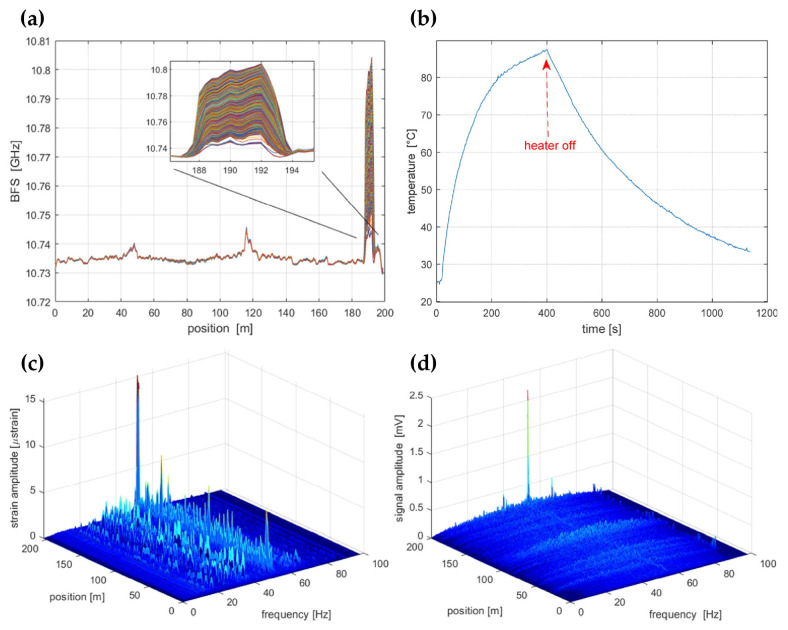
(**a**) BFS profiles when the fiber end is heated; (**b**) temperature evolution over the heated section; (**c**) vibration measured by BOTDA; (**d**) vibration measured by Φ-OTDR (reprinted from [192], under the Open Access Publishing Agreement from © 2021 Optical Society of America).

**Figure 26 sensors-23-07116-f026:**
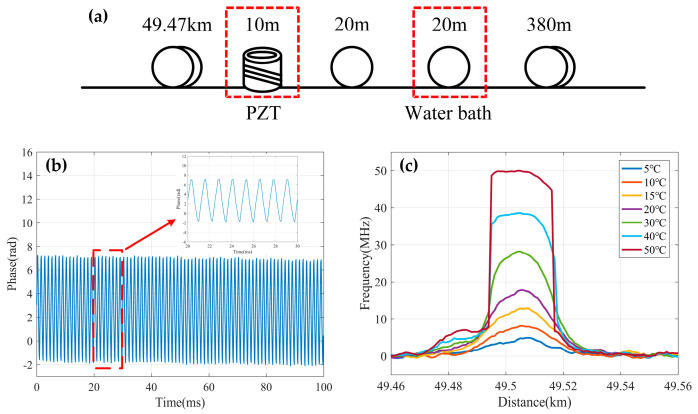
(**a**) Fiber arrangement in the test; (**b**) demodulated phase signal corresponding to an 800 Hz triangular vibration; (**c**) enlarged view of the hotspot at different heated temperatures (reprinted from [145], under the Open Access Publishing Agreement from © 2022 Optical Society of America).

**Figure 27 sensors-23-07116-f027:**
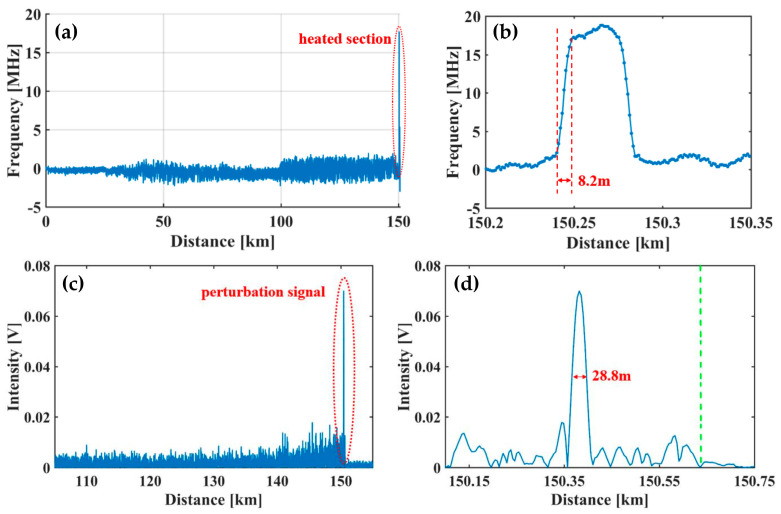
(**a**) BFS profiles along the whole fiber; (**b**) BFS profiles around the heated section; (**c**) demodulated Rayleigh signal along the whole fiber; (**d**) demodulated Rayleigh signal at the location of perturbation (reprinted from [194], under the terms and conditions of the Creative Commons Attribution License).

**Figure 28 sensors-23-07116-f028:**
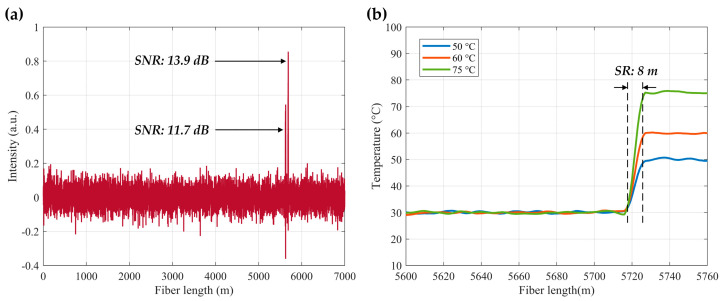
(**a**) Total of 885 superposed consecutive differential Φ-OTDR traces with intrusion applied to two fiber segments; (**b**) resolved temperature distribution with denoising method (adapted from [208], detailed curves are available in the reference paper).

**Figure 29 sensors-23-07116-f029:**
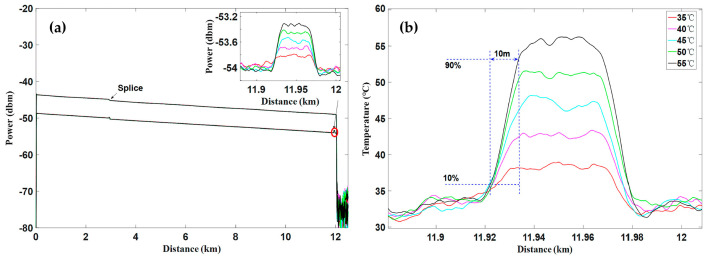
(**a**) Measured Raman Stokes and anti-Stokes traces along the whole fiber under various temperatures; (**b**) temperature distribution near the heated fiber section (reprinted from [206] with permission, © 2018 Elsevier).

**Figure 30 sensors-23-07116-f030:**
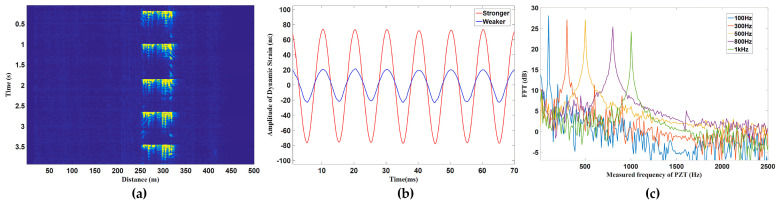
(**a**) Detected vibration waterfall regarding the knock event; (**b**) demodulated dynamic strain corresponding to the vibration exerting on the PZT; (**c**) frequency responses of the demodulated dynamic strain with different frequencies (reprinted from [206] with permission, © 2018 Elsevier).

**Figure 31 sensors-23-07116-f031:**
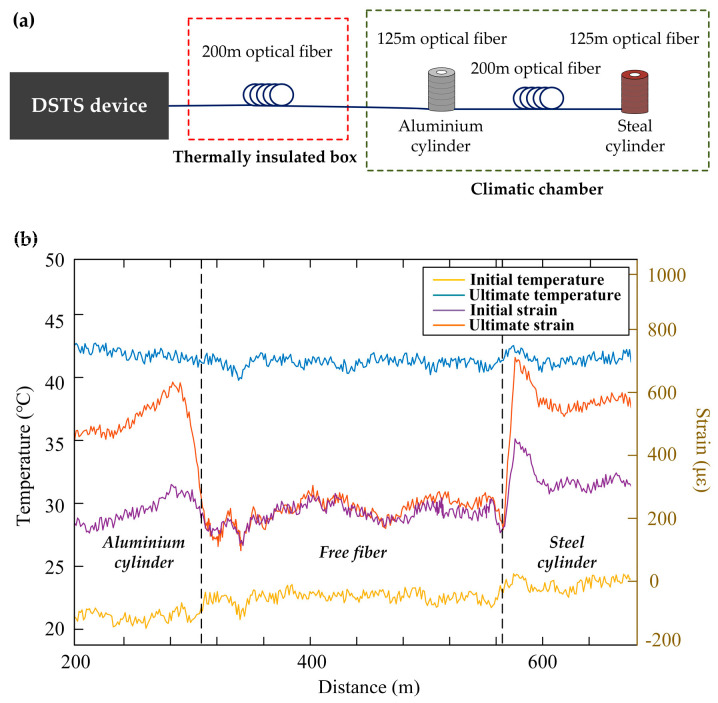
(**a**) Diagram of the test bench for independent measurement of temperature and strain; (**b**) independent measurement of distributed temperature and strain (adapted from [198], detailed curves are available in the reference paper).

**Figure 32 sensors-23-07116-f032:**
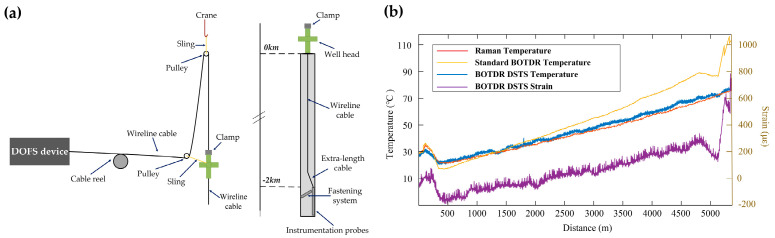
(**a**) Diagram of the good instrumentation for a distributed measurement of temperature and strain; (**b**) comparative curves of a distributed measurement among BOTDR DSTS (blue curve for temperature and purple curve for strain), standard ROTDR (red curve) and standalone BOTDR (yellow curve) (adapted from [198], detailed curves are available in the reference paper).

**Figure 33 sensors-23-07116-f033:**
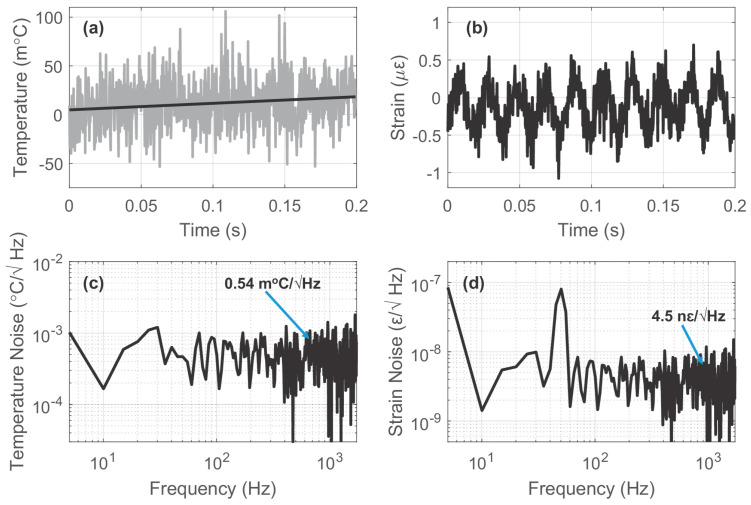
(**a**) Recovered temperature variation over time; (**b**) recovered strain variation over time; (**c**) amplitude spectral density curve of the recovered temperature; (**d**) amplitude spectral density curve of the recovered strain (reprinted from [172], under the Open Access Publishing Agreement from © 2023 Optica Publishing Group).

**Figure 34 sensors-23-07116-f034:**
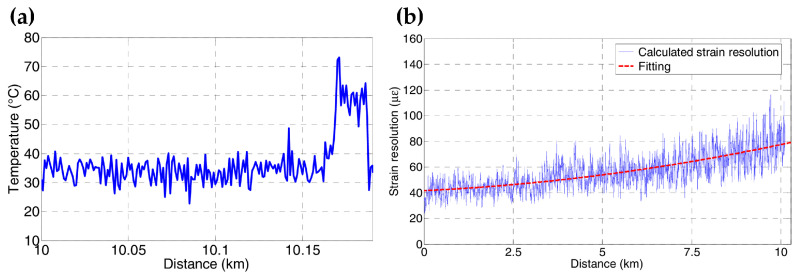
(**a**) Temperature distribution measured by Raman scattering; (**b**) strain resolution along the sensing fiber (reprinted from [210] with permission, © 2013 Optical Society of America).

**Figure 35 sensors-23-07116-f035:**
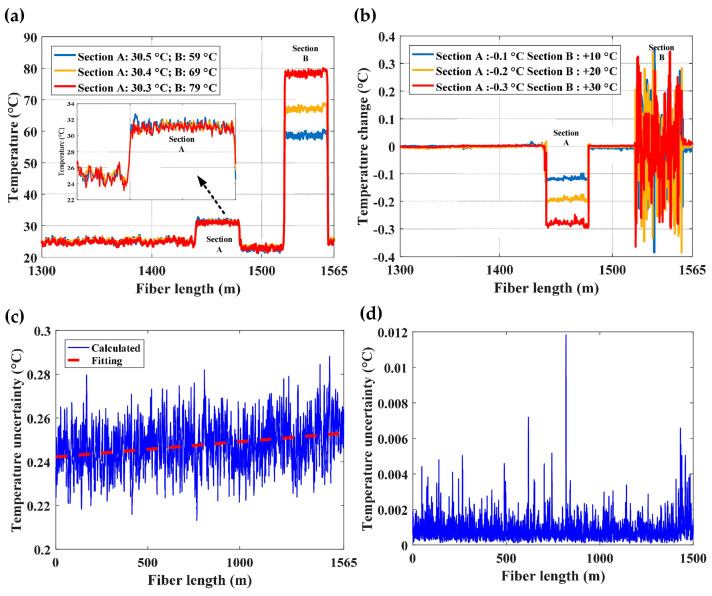
(**a**) Absolute temperature distribution measured by BOTDA; (**b**) relative temperature distribution measured by Φ-OTDR; (**c**) estimated temperature uncertainty distribution of BOTDA; (**d**) estimated temperature uncertainty distribution of Φ-OTDR (reprinted from [200], under the Open Access Publishing Agreement from © 2017 Optical Society of America).

**Figure 36 sensors-23-07116-f036:**
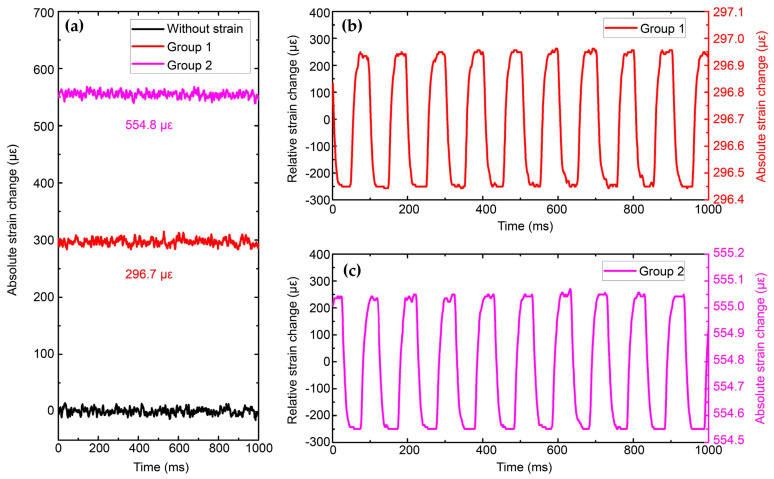
(**a**) Absolute strain change measured by BOTDA; (**b**) relative strain change of group 1 measured by Φ-OTDR; (**c**) relative strain change of group 2 measured by Φ-OTDR (reprinted from [199], under the terms and conditions of the Creative Commons Attribution License).

**Table 1 sensors-23-07116-t001:** Typical multi-parameter measurement performance based on the hybrid DOFS systems.

Classification	Subsystems Combination	Methods ^1^	Sensing Range	Spatial Resolution	Measurement Accuracy/Sensitivity ^2^	Ref.
Temperature/strain and vibration	POTDR/BOTDR	Orthogonal polarization multiplexing	4 km	10 m	0.2 MHz	[213]
	Φ-OTDR/BOTDR	Pulse modulation	10 km	80 cm for temperature/strain 3 m for vibration	<±1 MHz	[193]
	Φ-OTDR/BOTDA	Wavelength division multiplexing and distributed amplification	150.62 km	9 m for temperature/strain 30 m for vibration	±0.82 MHz	[194]
	Φ-OTDR/BOTDR	Frequency division multiplexing	49.9 km	20 m	0.381 MHz 1.235 nε/√Hz @100 Hz	[145]
Simultaneous temperature and strain	BOTDA/ROTDR	Wavelength division multiplexing and cyclic Simplex coding	10 km	1 m	2.6 °C/62 με	[223]
	COTDR/BOTDR	Frequency division multiplexing and coherent fading reduction	1 km/10 km optional	2 m	0.6 °C/20 με @1 km 3 °C/75 με @10 km	[198]
	FS-OTDR/BOTDA	Wavelength division multiplexing and enhanced slope-assisted method	500 m	5 m	16 m°C/140 nε 4.5 nε/√Hz @50 Hz	[172]
Temperature and vibration	Φ-OTDR/ROTDR	Space division multiplexing and wavelet transform denoising	5.76 km	8 m	0.5 °C	[208]
	Φ-OTDR/ROTDR	Wavelength division multiplexing and cyclic Simplex coding	5 km	5 m	<0.5 °C	[205]
	Φ-OTDR/ROTDR	Wavelength division multiplexing and heterodyne detection	12 km	10 m	0.95 °C	[206]
Comprehensive temperature measurement	FS-Φ-OTDR/BOTDA	Space division multiplexing	1.565 km	2.5 m	0.001 °C @Φ-OTDR 0.25 °C @BOTDA	[200]
Comprehensive strain measurement	FS-Φ-OTDR/BOTDA	Frequency-agile pulses	78 m	2 m	6.8 nε relative strain 5.4 με absolute strain	[199]
	FS-Φ-OTDR/BOTDA	Adaptive signal corrector	65 m	2 m	5.5 nε	[201]

^1^ Both the integration and the performance enhancement methods are included in the table. ^2^ Since the criterion of the provided results is not consistent among the reviewed works, here, the unit “MHz” represents the BFS measurement accuracy, the unit “°C” represents the temperature measurement accuracy or resolution, and the units with “ε” or “nε/√Hz” represent the strain measurement accuracy or sensitivity.

## Data Availability

Not applicable.

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
