# Peer review of "Hybrid Distributed Optical Fiber Sensor for the Multi-Parameter Measurements"

_sensors, 2023, doi:10.3390/s23167116_

Round 1

Reviewer 1 Report

In the manuscript of sensors-2526894, the authors summarized and discussed recent advances in the study of hybrid distributed optical fiber sensor for multi-parameter measurements, which is mainly based on the combination of the Rayleigh, Brillouin, and Raman scattering mechanisms, presenting the competence in measuring multi-parameter including temperature, strain, and vibration. The principle of each DOFS was summarized, and then the advances of hybrid DOFS systems are introduced, including the methods to combine the various DOFS techniques, and the optimization as well. The performance of multi-parameter measurements  related to the hybrid DOFSs was demonstrated and described in detail. The review is well-written and well-organized, reflecting the latest progress in this field, so I recommend it to be accepted after optional minor revision.

1. Could the ability of hybrid distributed optical fiber for humidity or gas sensing be discussed in the manuscript.

2. Offer a systematic table to compare the multi-parameter measurements of DOFS, including the key paramater, method, advantage and disadvantage.

Reviewer 2 Report

In this paper, the authors introduced the hybrid DOFS technology, which combines the Rayleigh, Brillouin, and Raman scattering mechanisms and integrates multiple DOFS systems in a single configuration, has attracted growing attention and been developed rapidly. This review highlights the latest progress of the hybrid DOFS technology for multi-parameter measurements. The basic principles of the light-scattering-based DOFSs are initially introduced, then the methods and sensing performances of various techniques are successively described. The challenges and prospects of the hybrid DOFS technology are discussed in the end, aiming to pave the way for a vaster range of applications. The idea behind this is interesting. However, I still have quite a number of concerns in this manuscript. There are times where there are not enough data to support the conclusions of the author. Please see some of the major concerns below.

1.The information for the sensor structure is not enough. The authors should give much more information about this. So the readers can get its reproducibility. For example -The typical configuration of BOTDR with coherent detection (figure 3) ? what is the laser type, which wavelengths ?.

Figure (5) what type of WDM system ? which wavelengths and what is the power penalty ?

2.  The authors should give much more information about the novelty of this paper, especially the effect of using this hybrid DOFS technology, which applications can be used this device?

3. More references need to be included in the introduction part to understand the applications of using Raman super-resolution signals:

a.     Super-resolved Raman spectroscopy- Spectroscopy Letters, 2013

b.     Super-resolved Raman spectra of toluene and toluene–chlorobenzene mixture- Spectroscopy Letters, 2015

c.     Improving Raman spectra of pure silicon using super-resolved method - Journal of Optics, 2019

4.  Much more discussion about the results should be given in this paper, especially the author needs to provide enough physicals mechanism analysis about the results.

Reviewer 3 Report

Current manuscript “Hybrid distributed optical fiber sensor for the multi-parameter Measurements” by “Zhou et al” summarizes and discusses recent advances in the study of hybrid distributed optical fiber sensor for multi-parameter measurements. The hybrid DOFS technology is mainly based on the combination of the Rayleigh, Brillouin, and Raman scattering mechanisms, presenting the competence in measuring multi-parameter including temperature, strain, and vibration. The review article may be accepted after addressing the following comments.

Ø  Summary of the reviewed articles is not provided at the end of every section

Ø  Conclusion section is not properly written.

Ø  Key findings of the performed study must be included in the revised version.

Ø  The Introduction should make a compelling case for why the study is useful along with a clear statement of its novelty or originality by providing relevant information and providing answers to basic questions such as: What is already known in the open literature? What is missing (i.e., research gaps)? What needs to be done, why and how?

Ø  Some potential articles were focused on the optical-sensing strategies. Authors should discuss those. Advances in optical-sensing strategies for the on-site detection of pesticides in agricultural foods. Emerging insights into the use of carbon-based nanomaterials for the electrochemical detection of heavy metal ions.

Ø  Clear statements of the novelty of the work should also appear briefly in the Abstract and Conclusions sections.

Ø  The manuscript has grammatical errors/ typos/ incomplete sentences and non-relative phrases.

Moderate editing of English language required

Round 2

Reviewer 2 Report

The new version can be published.

Reviewer 3 Report

Revised manuscript can be accepted for publication

 Minor editing of English language required